# TAda! Temporally-Adaptive Convolutions for Video Understanding

**Ziyuan Huang[1],   Shiwei Zhang[2,*],   Liang Pan[3],   Zhiwu Qing[2],**
**Mingqian Tang[2],   Ziwei Liu[3],   Marcelo H. Ang Jr[1,*]**
[1]Advanced Robotics Centre, National University of Singapore
[2]DAMO Academy, Alibaba Group   [3]S-Lab, Nanyang Technological University
`ziyuan.huang@u.nus.edu`, `{zhangjin.zsw,mingqian.tmq,qingzhiwu.qzw}@alibaba-inc.com`,
`{liang.pan, ziwei.liu}@ntu.edu.sg`, `mpeangh@nus.edu.sg`

## Abstract

Spatial convolutions[1] are widely used in numerous deep video models. It fundamentally assumes spatio-temporal invariance, *i.e.*, using shared weights for every location in different frames. This work presents **Temporally-Adaptive Convolutions (TAdaConv)** for video understanding[2], which shows that adaptive weight calibration along the temporal dimension is an efficient way to facilitate modelling complex temporal dynamics in videos. Specifically, TAdaConv empowers the spatial convolutions with temporal modelling abilities by calibrating the convolution weights for each frame according to its local and global temporal context. Compared to previous temporal modelling operations, TAdaConv is more efficient as it operates over the convolution kernels instead of the features, whose dimension is an order of magnitude smaller than the spatial resolutions. Further, the kernel calibration brings an increased model capacity. We construct TAda2D and TAda-ConvNeXt networks by replacing the 2D convolutions in ResNet and ConvNeXt with TAdaConv, which leads to at least on par or better performance compared to state-of-the-art approaches on multiple video action recognition and localization benchmarks. We also demonstrate that as a readily plug-in operation with negligible computation overhead, TAdaConv can effectively improve many existing video models with a convincing margin.

## 1 Introduction

Convolutions are an indispensable operation in modern deep vision models (He et al., 2016; Szegedy et al., 2015; Krizhevsky et al., 2012), whose different variants have driven the state-of-the-art performances of convolutional neural networks (CNNs) in many visual tasks (Xie et al., 2017; Dai et al., 2017; Zhou et al., 2019) and application scenarios (Howard et al., 2017; Yang et al., 2019). In the video paradigm, compared to the 3D convolutions (Tran et al., 2015), the combination of 2D spatial convolutions and 1D temporal convolutions are more widely preferred owing to its efficiency (Tran et al., 2018; Qiu et al., 2017). Nevertheless, 1D temporal convolutions still introduce non-negligible computation overhead on top of the spatial convolutions. Therefore, we seek to directly equip the spatial convolutions with temporal modelling abilities.

One essential property of the convolutions is the translation invariance (Ruderman & Bialek, 1994; Simoncelli & Olshausen, 2001), resulted from its local connectivity and shared weights. However, recent works in dynamic filtering have shown that strictly shard weights for all pixels may be suboptimal for modelling various spatial contents (Zhou et al., 2021; Wu et al., 2018).

Given the diverse nature of the temporal dynamics in videos, we hypothesize that the temporal modelling could also benefit from relaxed invariance along the temporal dimension. This means that the convolution weights for different time steps are no longer strictly shared. Existing dynamic filter networks can achieve this, but with two drawbacks. First, it is difficult for most of them (Zhou et al.,

---

[1]In this work, we use spatial convolutions and 2D convolutions interchangeably.
[2]Project page: `https://tadaconv-iclr2022.github.io/`.

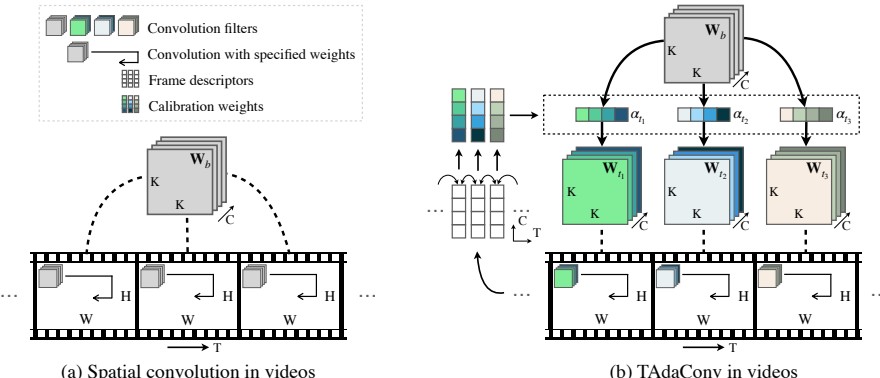

(a) Spatial convolution in videos   (b) TAdaConv in videos

Figure 1: **Comparisons between TAdaConv and the spatial convolutions in video models**. (a) Standard spatial convolutions in videos share the kernel weights between different frames. (b) Our TAdaConv adaptively calibrates the kernel weights for each frame by its temporal context.

2021; Yang et al., 2019) to leverage the existing pre-trained models such as ResNet. This is critical in video applications, since training video models from scratch is highly resource demanding (Feichtenhofer et al., 2019; Feichtenhofer, 2020) and prone to over-fitting on small datasets. Second, for most dynamic filters, the weights are generated with respect to its spatial context (Zhou et al., 2021; Jia et al., 2016) or the global descriptor (Chen et al., 2020a; Yang et al., 2019), which has difficulty in capturing the temporal variations between frames.

In this work, we present the Temporally-Adaptive Convolution (TAdaConv) for video understanding, where the convolution kernel weights are no longer fixed across different frames. Specifically, the convolution kernel for the $t$-th frame $\mathbf{W}_t$ is factorized to the multiplication of the base weight and a calibration weight: $\mathbf{W}_t = \boldsymbol{\alpha}_t \cdot \mathbf{W}_b$, where the calibration weight $\boldsymbol{\alpha}_t$ is adaptively generated from the input data for all channels in the base weight $\mathbf{W}_b$. For each frame, we generate the calibration weight based on the frame descriptors of its adjacent time steps as well as the global descriptor, which effectively encodes the local and global temporal dynamics in videos. The difference between TAdaConv and the spatial convolutions is visualized in Fig. 1.

The main advantages of this factorization are three-fold: **(i)** TAdaConv can be easily plugged into any existing models to enhance temporal modelling, and their pre-trained weights can still be exploited; **(ii)** the temporal modelling ability can be highly improved with the help of the temporally-adaptive weight; **(iii)** in comparison with temporal convolutions that often operate on the learned 2D feature maps, TAdaConv is more efficient by directly operating on the convolution kernels.

TAdaConv is proposed as a drop-in replacement for the spatial convolutions in existing models. It can both serve as a stand-alone temporal modelling module for 2D networks, or be inserted into existing convolutional video models to further enhance the ability to model temporal dynamics. For efficiency, we construct TAda2D by replacing the spatial convolutions in ResNet (He et al., 2016), which leads to at least on par or better performance than the state-of-the-arts. Used as an enhancement of existing video models, TAdaConv leads to notable improvements on multiple video datasets. The strong performance and the consistent improvements demonstrate that TAdaConv can be an important operation for modelling complex temporal dynamics in videos.

## 2 RELATED WORK

**ConvNets for temporal modelling.** A fundamental difference between videos and images lies in the temporal dimension, which makes temporal modeling an important research area for understanding videos. Recent deep CNNs for video understanding can be divided into two types. The first type jointly models spatio-temporal information by 3D convolutions (Carreira & Zisserman, 2017; Tran et al., 2015; Feichtenhofer, 2020; Tran et al., 2019). The second type builds upon 2D networks, where most approaches employ 2D convolutions that share weights among all the frames for spatial modelling, and design additional operations for temporal modelling, such as temporal shift (Lin et al., 2019a), temporal difference (Wang et al., 2021; Jiang et al., 2019), temporal convolution (Tran et al., 2018; Liu et al., 2021b) and correlation operation (Wang et al., 2020), *etc*. Our work directly empowers the spatial convolutions with temporal modelling abilities, which can be further coupled with other temporal modelling operations for stronger video recognition performances.

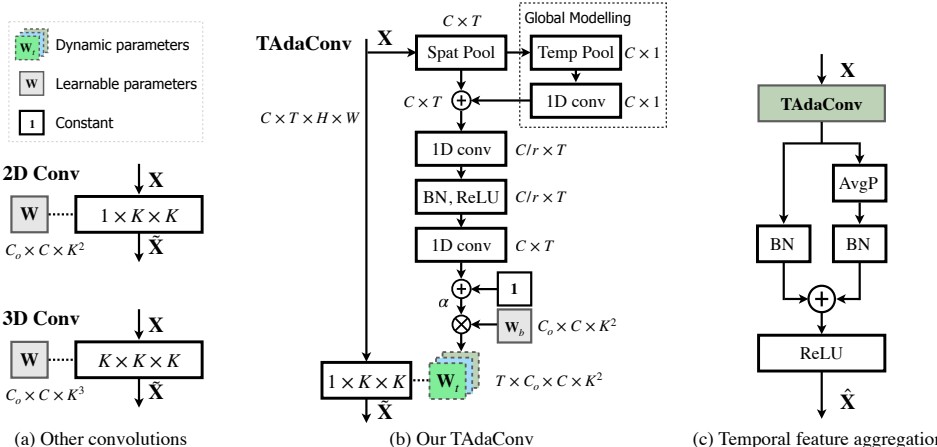

(a) Other convolutions          (b) Our TAdaConv          (c) Temporal feature aggregation

Figure 2: **An instantiation of TAdaConv and the temporal feature aggregation used in TAda2D.** (a) Standard convolutions used in video models. (b) Our TAdaConv using non-linear weight calibrations with global temporal context. (c) The temporal feature aggregation scheme used in TAda2D.

**Dynamic networks.** Dynamic networks refer to networks with content-adaptive weights or modules, such as dynamic filters/convolutions (Jia et al., 2016; Yang et al., 2019; Li et al., 2021), dynamic activations (Li et al., 2020d; Chen et al., 2020b), and dynamic routing (Wang et al., 2018; Li et al., 2020c), *etc*. Dynamic networks have demonstrated their exceeding network capacity and thus performance compared to the static ones. In video understanding, dynamic filter (Liu et al., 2021b) or temporal aggregation (Meng et al., 2021) also demonstrate strong capability of temporal modelling. Recently, some convolutions applies spatially-adaptive weights (Elsayed et al., 2020; Chen et al., 2021), showing the benefit of relaxing the spatial invariance in modelling diverse visual contents. Similarly, our proposed TAdaConv enhances temporal modelling by relaxing the invariance along the temporal dimension. TAdaConv has two key differences from previous works: (i) the convolution weight in TAdaConv is factorized into a base weight and a calibration weight, which enables TAdaConv to exploit pre-trained weights; (ii) the calibration weights are generated according to the temporal contexts.

## 3 TAdaConv: Temporally-adaptive Convolutions

In this work, we seek to empower the spatial convolutions with temporal modelling abilities. Inspired by the calibration process of temporal convolutions (Sec. 3.1), TAdaConv dynamically calibrates the convolution weights for each frame according to its temporal context (Sec. 3.2).

### 3.1 Revisiting temporal convolutions

We first revisit the temporal convolutions to showcase its underlying process and its relation to dynamic filters. We consider depth-wise temporal convolution for simplicity, which is more widely used because of its efficiency (Liu et al., 2021b; Jiang et al., 2019). Formally, for a $3 \times 1 \times 1$ temporal convolution filter parameterized by $\boldsymbol{\beta} = [\boldsymbol{\beta}_1, \boldsymbol{\beta}_2, \boldsymbol{\beta}_3]$ and placed (ignoring normalizations) after the 2D convolution parameterized by $\mathbf{W}$, the output feature $\tilde{\mathbf{x}}_t$ of the $t$-th frame can be obtained by:

$$\tilde{\mathbf{x}}_t = \boldsymbol{\beta}_1 \cdot \delta(\mathbf{W} * \mathbf{x}_{t-1}) + \boldsymbol{\beta}_2 \cdot \delta(\mathbf{W} * \mathbf{x}_t) + \boldsymbol{\beta}_3 \cdot \delta(\mathbf{W} * \mathbf{x}_{t+1}) , \qquad (1)$$

where the $\cdot$ indicates the element-wise multiplication, $*$ denotes the convolution over the spatial dimension and $\delta$ denotes ReLU activation (Nair & Hinton, 2010). It can be rewritten as follows:

$$\tilde{\mathbf{x}}_t = \mathbf{W}_{t-1} * \mathbf{x}_{t-1} + \mathbf{W}_t * \mathbf{x}_t + \mathbf{W}_{t+1} * \mathbf{x}_{t+1} , \qquad (2)$$

where $\mathbf{W}_{t-1}^{i,j} = \mathbf{M}_{t-1}^{i,j} \cdot \boldsymbol{\beta}_1 \cdot \mathbf{W}, \mathbf{W}_t^{i,j} = \mathbf{M}_t^{i,j} \cdot \boldsymbol{\beta}_2 \cdot \mathbf{W}$ and $\mathbf{W}_{t+1}^{i,j} = \mathbf{M}_{t+1}^{i,j} \cdot \boldsymbol{\beta}_3 \cdot \mathbf{W}$ are spatio-temporal location adaptive convolution weights. $\mathbf{M}_t \in \mathbb{R}^{C \times H \times W}$ is a dynamic tensor, with its value dependent on the result of the spatial convolutions (see Appendix A for details). Hence, the temporal convolutions in the (2+1)D convolution essentially performs **(i)** weight calibration on the spatial convolutions and **(ii)** feature aggregation between adjacent frames. However, if the temporal modelling is achieved by coupling temporal convolutions to spatial convolutions, a non-negligible computation overhead is still introduced (see Table 2).

## 3.2 FORMULATION OF TADACONV

For efficiency, we set out to directly empower the spatial convolutions with temporal modelling abilities. Inspired by the recent finding that the relaxation of spatial invariance strengthens spatial modelling (Zhou et al., 2021; Elsayed et al., 2020), we hypothesize that temporally adaptive weights can also help temporal modelling. Therefore, the convolution weights in a TAdaConv layer are varied on a frame-by-frame basis. Since we observe that previous dynamic filters can hardly utilize the pretrained weights, we take inspiration from our observation in the temporal convolutions and factorize the weights for the $t$-th frame $\mathbf{W}_t$ into the multiplication of a base weight $\mathbf{W}_b$ shared for all frames, and a calibration weight $\boldsymbol{\alpha}_t$ that are different for each time step:

$$\tilde{\mathbf{x}}_t = \mathbf{W}_t * \mathbf{x}_t = (\boldsymbol{\alpha}_t \cdot \mathbf{W}_b) * \mathbf{x}_t , \tag{3}$$

**Calibration weight generation.** To allow for the TAdaConv to model temporal dynamics, it is crucial that the calibration weight $\boldsymbol{\alpha}_t$ for the $t$-th frame takes into account not only the current frame, but more importantly, its temporal context, *i.e.,* $\boldsymbol{\alpha}_t = \mathcal{G}(..., \mathbf{x}_{t-1}, \mathbf{x}_t, \mathbf{x}_{t+1}, ...)$. Otherwise TAdaConv would degenerate to a set of unrelated spatial convolutions with different weights applied on different frames. We show an instantiation of the generation function $\mathcal{G}$ in Fig. 2(b).

In our design, we aim for efficiency and the ability to capture inter-frame temporal dynamics. For efficiency, we operate on the frame description vectors $\mathbf{v} \in \mathbb{R}^{T \times C}$ obtained by the global average pooling over the spatial dimension $\mathrm{GAP}_s$ for each frame, *i.e.,* $\mathbf{v}_t = \mathrm{GAP}_s(\mathbf{x}_t)$. For temporal modelling, we apply stacked two-layer 1D convolutions $\mathcal{F}$ with a dimension reduction ratio of $r$ on the local temporal context $\mathbf{v}_t^{adj} = \{\mathbf{v}_{t-1}, \mathbf{v}_t, \mathbf{v}_{t+1}\}$ obtained from $\mathbf{x}_t^{adj} = \{\mathbf{x}_{t-1}, \mathbf{x}_t, \mathbf{x}_{t+1}\}$:

$$\mathcal{F}(\mathbf{x}_t^{adj}) = \mathrm{Conv1D}^{C/r \to C}(\delta(\mathrm{BN}(\mathrm{Conv1D}^{C \to C/r}(\mathbf{v}_t^{adj})))) . \tag{4}$$

where $\delta$ and BN denote the ReLU (Nair & Hinton, 2010) and batchnorm (Ioffe & Szegedy, 2015).

In order for a larger inter-frame field of view in complement to the local 1D convolution, we further incorporate global temporal information by adding a global descriptor $\mathbf{g}$ to the weight generation process $\mathcal{F}$ through a linear mapping function FC:

$$\mathcal{F}(\mathbf{x}_t^{adj}, \mathbf{g}) = \mathrm{Conv1D}^{C/r \to C}(\delta(\mathrm{BN}(\mathrm{Conv1D}^{C \to C/r}(\mathbf{v}_t^{adj} + \mathrm{FC}^{C \to C}(\mathbf{g}))))) , \tag{5}$$

where $\mathbf{g} = \mathrm{GAP}_{st}(\mathbf{x})$ with $\mathrm{GAP}_{st}$ being global average pooling on spatial and temporal dimensions.

**Initialization.** The TAdaConv is designed to be readily inserted into existing models by simply replacing the 2D convolutions. For an effective use of the pre-trained weights, TAdaConv is initialized to behave exactly the same as the standard convolution. This is achieved by zero-initializing the weight of the last convolution in $\mathcal{F}$ and adding a constant vector $\mathbf{1}$ to the formulation:

$$\boldsymbol{\alpha}_t = \mathcal{G}(\mathbf{x}) = \mathbf{1} + \mathcal{F}(\mathbf{x}_t^{adj}, \mathbf{g}) . \tag{6}$$

In this way, at initial state, $\mathbf{W}_t = \mathbf{1} \cdot \mathbf{W}_b = \mathbf{W}_b$, where we load $\mathbf{W}_b$ with the pre-trained weights.

**Calibration dimension.** The base weight $\mathbf{W}_b \in \mathbb{R}^{C_{\mathrm{out}} \times C_{\mathrm{in}} \times k^2}$ can be calibrated in different dimensions. We instantiate the calibration on the $C_{in}$ dimension ($\boldsymbol{\alpha}_t \in \mathbb{R}^{1 \times C_{in} \times 1}$), as the weight generation based on the input features yields a more precise estimation for the relation of the input channels than the output channels or spatial structures (empirical analysis in Table 7).

**Comparison with other dynamic filters.** Table 1 compares TAdaConv with existing dynamic filters. Mixtue-of-experts based dynamic filters such as CondConv dynamically aggregates multiple kernels to generate the weights that are shared for all locations. The weights in most other dynamic filters are completely generated from the input, such as DynamicFilter (Jia et al., 2016) and

Table 1: Comparison with other dynamic filters.

| Operations | Temporal modelling | Location adaptive | Pretrained weights |
|---|---|---|---|
| CondConv | ✗ | ✗ | ✗ |
| DynamicFilter | ✗ | ✗ | ✗ |
| DDF | ✗ | ✓ | ✗ |
| TAM | ✓ | ✗ | ✗ |
| TAdaConv | ✓ | ✓ | ✓ |

DDF (Zhou et al., 2021) in images and TAM (Liu et al., 2021b) in videos. Compared to image based ones, TAdaConv achieves temporal modelling by generating weights from the local and global temporal context. Compared to TANet (Liu et al., 2021b), TAdaConv is better at temopral modellling because of temporally adaptive weights. Further, TAdaConv can effectively generate weights identical to the pre-trained ones, while it is difficult for previous approaches to exploit pre-trained models. More detailed comparisons of dynamic filters are included in Appendix J.

Table 2: Comparison of (2+1)D convolution and TAdaConv in FLOPs and number of parameters. Example setting for operation: $C_o = C_i = 64$, $k = 3$, $T = 8$, $H = W = 56$ and $r = 4$. Example setting for network: ResNet-50 with input resolution $8 \times 224^2$. Colored numbers denote the extra FLOPs/Params. added to 2D convolutions or ResNet-50. Refer to Appendix D for model structures.

| | (2+1)D Conv | TAdaConv |
|---|---|---|
| FLOPs | $C_o \times C_i \times k^2 \times THW$ $+C_o \times C_i \times k \times THW$ | $C_o \times C_i \times k^2 \times THW + C_i \times (THW + T)$ $+C_i \times C_i/r \times (2 \times k \times T + 1) + C_o \times C_i \times k^2 \times T$ |
| E.G. Op | 1.2331 (+0.308, ↑33%) | 0.9268 (+0.002, ↑0.2%) |
| E.G. Net | 37.94 (+4.94, ↑15%) | 33.02 (+0.02, ↑0.06%) |
| Params. | $C_o \times C_i \times k^2 + C_o \times C_i \times k$ | $C_o \times C_i \times k^2 + 2 \times C_i \times C_i/r \times k$ |
| E.G. Op. | 49,152 (+12,288, ↑33%) | 43,008 (+6,144, ↑17%) |
| E.G. Net | 28.1M (+3.8M, ↑15.6%) | 27.5M (+3.2M, ↑13.1%) |

**Comparison with temporal convolutions.** Table 2 compares the TAdaConv with R(2+1)D in parameters and FLOPs, which shows most of our additional computation overhead on top of the spatial convolution is an order of magnitude less than the temporal convolution. For detailed computation analysis and comparison with other temporal modelling approaches, please refer to Appendix B.

## 4   TADA2D: TEMPORALLY ADAPTIVE 2D NETWORKS

We construct TAda2D networks by replacing the 2D convolutions in ResNet (R2D, see Appendix D) with our proposed TAdaConv. Additionally, based on strided average pooling, we propose a temporal feature aggregation module for the 2D networks, corresponding to the second essential step for the temporal convolutions. As illustrated in Fig. 2(c), the aggregation module is placed after TAdaConv. Formally, given the output of TAdaConv $\tilde{\mathbf{x}}$, the aggregated feature can be obtained as follows:

$$\mathbf{x}_{aggr} = \delta(\text{BN}_1(\tilde{\mathbf{x}}) + \text{BN}_2(\text{TempAvgPool}_k(\tilde{\mathbf{x}}))) , \qquad (7)$$

where $\text{TempAvgPool}_k$ denotes strided temporal average pooling with kernel size of $k$. We use different batch normalization parameters for the features extracted by TAdaConv $\tilde{\mathbf{x}}$ and aggregated by strided average pooling $\text{TempAvgPool}_k(\tilde{\mathbf{x}})$, as their distributions are essentially different. During initialization, we load pre-trained weights to $\text{BN}_1$, and initialize the parameters of $\text{BN}_2$ to zero. Coupled with the initialization of TAdaConv, the initial state of the TAda2D is exactly the same as the Temporal Segment Networks (Wang et al., 2016), while the calibration and the aggregation notably increases the model capacity with training (See Appendix I). In the experiments, we refer to this structure as the shortcut (Sc.) branch and the separate BN (SepBN.) branch.

## 5   EXPERIMENTS ON VIDEO CLASSIFICATION

To show the effectiveness and generality of the proposed approach, we present comprehensive evaluation of TAdaConv and TAda2D on two video understanding tasks using four large-scale datasets.

**Datasets.** For video classification, we use Kinetics-400 (Kay et al., 2017), Something-Something-V2 (Goyal et al., 2017), and Epic-Kitchens-100 (Damen et al., 2020). *K400* is a widely used action classification dataset with 400 categories covered by ~300K videos. *SSV2* includes 220K videos with challenging spatio-temporal interactions in 174 classes. *EK100* includes 90K segments labelled by 97 verb and 300 noun classes with actions defined by the combination of nouns and verbs. For action localization, we use HACS (Zhao et al., 2019) and Epic-Kitchens-100 (Damen et al., 2020).

**Model.** In our experiments, we mainly use ResNet (R2D) as our base model, and construct TAda2D by replacing the spatial convolutions with the TAda-structure in Fig. 2(c). Alternatively, we also construct TAdaConvNeXt based on the recent ConvNeXt model (Liu et al., 2022). For TAdaConvNeXt, we use a tubelet embedding stem similar to (Arnab et al., 2021) and only use TAdaConv to replace the depth-wise convolutions in the model. More details are included in Appendix D.

**Training and evaluation.** During training, 8, 16 or 32 frames are sampled with temporal jittering, following convention (Lin et al., 2019a; Liu et al., 2021b; Feichtenhofer et al., 2019). We include further training details in the appendix C. For evaluation, we use three spatial crops with 10 or 4 clips (K400&EK100), or 2 clips (SSV2) uniformly sampled along the temporal dimension. Each crop has the size of 256×256, which is obtained from a video with its shorter side resized to 256.

Table 3: Plug-in evaluation of TAdaConv in existing video models on K400 and SSV2 datasets.

| Base Model | TAdaConv | Frames | Params. | GFLOPs | K400 | Δ | SSV2 | Δ |
|---|---|---|---|---|---|---|---|---|
| SlowOnly 8×8⋆ | ✗ | 8 | 32.5M | 54.52 | 74.56 | - | 60.31 | - |
|  | ✓ | 8 | 35.6M | 54.53 | 75.85 | **+1.29** | 63.30 | **+2.99** |
| SlowFast 4×16⋆ | ✗ | 4+32 | 34.5M | 36.10 | 75.03 | - | 56.71 | - |
|  | ✓ | 4+32 | 37.7M | 36.11 | 76.47 | **+1.44** | 59.80 | **+3.09** |
| SlowFast 8×8⋆ | ✗ | 8+32 | 34.5M | 65.71 | 76.19 | - | 61.54 | - |
|  | ✓ | 8+32 | 37.7M | 65.73 | 77.43 | **+1.24** | 63.88 | **+2.34** |
| R(2+1)D⋆ | ✗ | 8 | 28.1M | 49.55 | 73.63 | - | 61.06 | - |
|  | ✓(2d) | 8 | 31.2M | 49.57 | 75.19 | **+1.56** | 62.86 | **+1.80** |
|  | ✓(2d+1d) | 8 | 34.4M | 49.58 | 75.36 | **+1.73** | 63.78 | **+2.72** |
| R3D⋆ | ✗ | 8 | 47.0M | 84.23 | 73.83 | - | 59.86 | - |
|  | ✓(3d) | 8 | 50.1M | 84.24 | 74.91 | **+1.08** | 62.85 | **+2.99** |

Notation ⋆ indicates our own implementation. See Appendix D for details on the model structure.

Table 4: Calibration weight generation. *K:* kernel size; *Lin./Non-Lin.:* linear/non-linear weight generation; *G:* global information **g**.

| Model | TAdaConv | K. | G. | Top-1 |
|---|---|---|---|---|
| TSN⋆ | - | - | - | 32.0 |
| Ours | Lin. | 1 | ✗ | 37.5 |
|  | Lin. | 3 | ✗ | 56.5 |
|  | Non-Lin. | (1, 1) | ✗ | 36.8 |
|  | Non-Lin. | (3, 1) | ✗ | 57.1 |
|  | Non-Lin. | (1, 3) | ✗ | 57.3 |
|  | Non-Lin. | (3, 3) | ✗ | 57.8 |
|  | Lin. | 1 | ✓ | 53.4 |
|  | Non-Lin. | (1, 1) | ✓ | 54.4 |
|  | Non-Lin. | (3, 3) | ✓ | 59.2 |

Table 5: Feature aggregation scheme. *FA:* feature aggregation; *Sc:* shortcut for convolution feature; *SepBN:* separate batch norm.

| TAdaConv | FA. | Sc. | SepBN. | Top-1 | Δ |
|---|---|---|---|---|---|
| ✗ | - | - | - | 32.0 | - |
| ✓ | - | - | - | 59.2 | +27.2 |
| ✗ | Avg. | ✗ | - | 47.9 | +15.9 |
| ✗ | Avg. | ✓ | ✗ | 49.0 | +17.0 |
| ✗ | Avg. | ✓ | ✓ | 57.0 | +25.0 |
| ✓ | Avg. | ✗ | - | 60.1 | +28.1 |
| ✓ | Avg. | ✓ | ✗ | 61.5 | +29.5 |
| ✓ | Avg. | ✓ | ✓ | 63.8 | **+31.8** |
| ✓ | Max. | ✓ | ✓ | 63.5 | +31.5 |
| ✓ | Mix. | ✓ | ✓ | 63.7 | +31.7 |

## 5.1 TAdaConv on existing video backbones

TAdaConv is designed as a plug-in substitution for the spatial convolutions in the video models. Hence, we first present plug-in evaluations in Table 3. TAdaConv improves the classification performance with negligible computation overhead on a wide range of video models, including Slow-Fast (Feichtenhofer et al., 2019), R3D (Hara et al., 2018) and R(2+1)D (Tran et al., 2018), by an average of 1.3% and 2.8% respectively on K400 and SSV2 at an extra computational cost of less than 0.02 GFlops. Further, not only can TAdaConv improve spatial convolutions, it also notably improve 3D and 1D convolutions. For fair comparison, all models are trained using the same training strategy. Further plug-in evaluations for action classification is presented in Appendix G.

## 5.2 Ablation studies

We present thorough ablation studies for the justification of our design choices and the effectiveness of our TAdaConv in modelling temporal dynamics. SSV2 is used as the evaluation benchmark, as it is widely acknowledged to have more complex spatio-temporal interactions.

**Dynamic vs. learnable calibration.** We first compare different source of calibration weights (with our initialization strategy) in Table 6. We compare our calibration with no calibration, calibration using learnable weights, and calibration using dynamic weights generated only from a global descriptor (C×1). Compared with the baseline (TSN (Wang et al., 2016)) with no calibration, learnable calibration with shared weights has limited improvement, while temporally varying learnable cal-

Table 6: Benefit of dynamic calibration. *T.V.:* temporally varying. *: w/o our init.

| Calibration | T.V. | Top-1 | Top-1* |
|---|---|---|---|
| None | ✗ | - | 32.0 |
| Learnable | ✗ | 34.3 | 32.6 |
|  | ✓ | 45.4 | 43.8 |
| Dynamic | ✗ | 51.2 | 41.7 |
|  | ✓ | 53.8 | 49.8 |
| TAda | ✓ | 59.2 | 47.8 |

ibration (different calibration weights for different temporal locations) performs much stronger. A larger improvement is observed when we use dynamic calibration, where temporally varying calibration further raises the accuracy. The results also validate our hypothesis that temporal modelling can benefit from temporally adaptive weights. Further, TAdaConv generates calibration weight from both local (C×T) and global (C×1) contexts and achieves the highest performance.

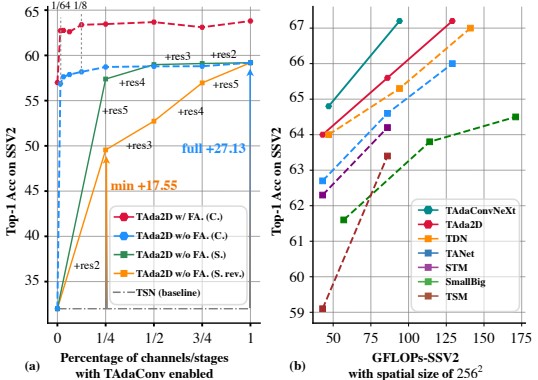 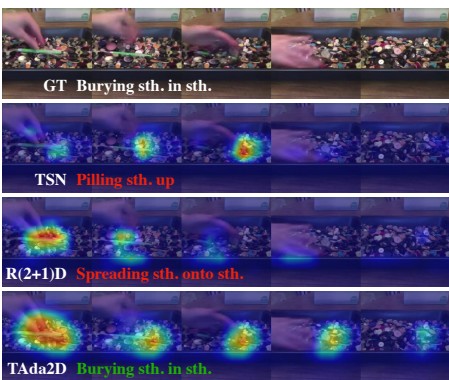

Figure 3: The classification performance of TAda2D (a) with different channels (C.) and stages (S.) enabled; (b) in comparison with other state-of-the-arts.

Figure 4: Grad-CAM visualization and prediction comparison between TSN, R(2+1)D and TAda2D (more examples in Fig. A3).

**Calibration weight initialization.** Next, we show that our initialization strategy for the calibration weight generation plays a critical role for dynamic weight calibration. As in Table 6, randomly initializing learnable weights slightly degrades the performance, while randomly initializing dynamic calibration weights (by randomly initializing the last layer of the weight generation function) notably degenerates the performance. It is likely that randomly initialized dynamic calibration weights purturb the pre-trained weights more severely than the learnable weights since it is dependent on the input. Further comparisons on the initialization are shown in the Table A7 in the Appendix.

**Calibration weight generation function.** Having established that the temporally adaptive dynamic calibration with appropriate initialization can be an ideal strategy for temporal modelling, we further ablate different ways for generating the calibration weight in Table 4. Linear weight generation function (*Lin.*) applies a single 1D convolution to generate the calibration weight, while non-linear one (*Non-Lin.*) uses two stacked 1D convolutions with batch normalizations and ReLU activation in between. When no temporal context is considered (K.=1 or (1,1)), TAdaConv can still improve the baseline but with a limited gap. Enlarging the kernel size to cover the temporal context (K.=3, (1,3), (3,1) or (3,3)) further yields a boost of over 20% on the accuracy, with K.=(3,3) having the strongest performance. This shows the importance of the local temporal context during calibration weight generation. Finally, for the scope of temporal context, introducing global context to frame descriptors performs similarly to only generating temporally adaptive calibration weights solely on the global context (in Table 6). The combination of the global and temporal context yields a better performance for both variants. We further show in Appendix J that this function in our TAdaConv yields a better calibration on the base weight than other existing dynamic filters.

**Feature aggregation.** The feature aggregation module used in the TAda2D network is ablated in Table 5. First, the performance is similar for plain aggregation $\mathbf{x} = \text{Avg}(\mathbf{x})$ and aggregation with a shortcut (Sc.) branch $\mathbf{x} = \mathbf{x} + \text{Avg}(\mathbf{x})$, with Sc. being slightly better. Separating the batchnorm (Eq. 7) for the shortcut and the aggregation branch brings notable improvement. Strided max and mix (avg+max) pooling slightly underperform the average pooling variant. Overall, the combination of TAdaConv and our feature aggregation scheme has an advantage over the TSN baseline of 31.8%.

**Calibration dimension.** Multiple dimensions can be calibrated in the base weight. Table 7 shows that calibrating the channel dimension more suitable than the spatial dimension, which means that the spatial structure of the original convolution kernel should be retained. Within channels, the calibration works better on $C_{\text{in}}$ than $C_{\text{out}}$ or both combined. This is probably because the calibration weight generated by the input feature can better adapt to itself.

Table 7: Calibration dimension.

| Cal. dim. | $\Delta_{\text{Parms.}}$ | $\Delta_{\text{GFLOPs}}$ | Top-1 |
|---|---|---|---|
| $C_{\text{in}}$ | 3.16M | 0.016 | 63.8 |
| $C_{\text{out}}$ | 3.16M | 0.016 | 63.4 |
| $C_{\text{in}} \times C_{\text{out}}$ | 4.10M | 0.024 | 63.7 |
| $k^2$ | 2.24M | 0.009 | 62.7 |

**Different stages employing TAdaConv.** The solid lines in Fig 3 show the stage by stage replacement of the spatial convolutions in a ResNet model. It has a minimum improvement of 17.55%, when TAdaConv is employed in *Res2*. Compared to early stages, later stages contribute more to the final performance, as later stages provide more accurate calibration because of its high abstraction level. Overall, TAdaConv is used in all stages for the highest accuracy.

Table 8: Comparison with the top approaches on Something-Something-V2 (Goyal et al., 2017).

| Model | Backbone | Frames×clips×crops | GFLOPs | Top-1 | Top-5 |
|---|---|---|---|---|---|
| TDN (Wang et al., 2021) | ResNet-50 | $(8f+32f)\times1\times1$ | 47 | 64.0 | 88.8 |
| TDN (Wang et al., 2021) | ResNet-50 | $(16f+64f)\times1\times1$ | 94 | 65.3 | 89.5 |
| TDN (Wang et al., 2021) | ResNet-50 | $(8f+32f+16f+64f)\times1\times1$ | 141 | 67.0 | 90.3 |
| TSM (Lin et al., 2019a) | ResNet-50 | $8f\times2\times3$ | 43 | 59.1 | 85.6 |
| TSM (Lin et al., 2019a) | ResNet-50 | $16f\times2\times3$ | 86 | 63.4 | 88.5 |
| SmallBigNet (Li et al., 2020a) | ResNet-50 | $8f\times2\times3$ | 57 | 61.6 | 87.7 |
| SmallBigNet (Li et al., 2020a) | ResNet-50 | $16f\times2\times3$ | 114 | 63.8 | 88.9 |
| SmallBigNet (Li et al., 2020a) | ResNet-50 | $(8f+16f)\times2\times3$ | 171 | 64.5 | 89.1 |
| TANet (Liu et al., 2021b) | ResNet-50 | $8f\times2\times3$ | 43 | 62.7 | 88.0 |
| TANet (Liu et al., 2021b) | ResNet-50 | $16f\times2\times3$ | 86 | 64.6 | 89.5 |
| TANet (Liu et al., 2021b) | ResNet-50 | $(8f+16f)\times2\times3$ | 129 | 64.6 | 89.5 |
| TAda2D (Ours) | ResNet-50 | $8f\times2\times3$ | 43 | 64.0 | 88.0 |
| TAda2D (Ours) | ResNet-50 | $16f\times2\times3$ | 86 | 65.6 | 89.2 |
| TAda2D$_{En}$ (Ours) | ResNet-50 | $(8f+16f)\times2\times3$ | 129 | 67.2 | 89.8 |
| TAdaConvNeXt-T (Ours) | ConvNeXt-T | $16f\times2\times3$ | 47 | 64.8 | 88.8 |
| TAdaConvNeXt-T (Ours) | ConvNeXt-T | $32f\times2\times3$ | 94 | 67.1 | 90.4 |

Gray font indicates models with different inputs. FLOPs are calculated with 256×256 resolution as in the evaluation.

Table 9: Comparison with the state-of-the-art approaches over action classification on Epic-Kitchens-100 (Damen et al., 2020). ⋆ indicates our own implementation for fair comparison. ↑ indicates the main evaluation metric for the dataset.

| Model | Frames | Top-1 | | | Top-5 | | |
|---|---|---|---|---|---|---|---|
| | | Act.↑ | Verb | Noun | Act.↑ | Verb | Noun |
| TSN (Wang et al., 2016) | 8 | 33.19 | 60.18 | 46.03 | 55.13 | 89.59 | 72.90 |
| TRN (Zhou et al., 2018) | 8 | 35.34 | 65.88 | 45.43 | 56.74 | 90.42 | 71.88 |
| TSM (Lin et al., 2019a) | 8 | 38.27 | 67.86 | 49.01 | 60.41 | 90.98 | 74.97 |
| SlowFast (Feichtenhofer et al., 2019) | 8+32 | 38.54 | 65.56 | 50.02 | 58.60 | 90.00 | 75.62 |
| TSN⋆ (Our baseline) | 8 | 30.15 | 51.89 | 45.77 | 53.00 | 87.51 | 72.16 |
| TAda2D (Ours) | 8 | 41.61 | 65.14 | 52.39 | 61.98 | 90.54 | 76.45 |

**Different proportion of channels calibrated by TAdaConv.** Here, we calibrate only a proportion of channels using TAdaConv and leave the other channels uncalibrated. The results are presented as dotted lines in Fig 3. We find TAdaConv can improve the baseline by a large margin even if only 1/64 channels are calibrated, with larger proportion yielding further larger improvements.

**Visualizations.** We qualitatively evaluate our approach in comparison with the baseline approaches (TSN and R(2+1)D) by presenting the Grad-CAM (Selvaraju et al., 2017) visualizations of the last stage in Fig. 4. TAda2D can more completely spot the key information in videos, thanks to the temporal reasoning based on global spatial information and the global temporal information.

## 5.3 MAIN RESULTS

**SSV2.** As shown in Table 8, TAda2D outperforms previous approaches using the same number of frames. Compared to TDN that uses more frames, TAda2D performs competitively. Visualization in Fig. 3(b) also demonstrates the superiority of our performance/efficiency trade-off. An even stronger performance is achieved with a similar amount of computation by TAdaConvNeXt, which provides an accuracy of 67.1% with 94GFLOPs.

**Epic-Kitchens-100.** Table 9 lists our results on EK100 in comparison with the previous approaches[3]. We calculate the final action prediction following the strategies in Huang et al. (2021). For fair comparison, we reimplemented our baseline TSN using the same training and evaluation strategies. TAda2D improves this baseline by 11.46% on the action prediction. Over previous approaches, TAda2D achieves a higher accuracy with a notable margin.

**Kinetics-400.** Comparison with the state-of-the-art models on Kinetics-400 is presented in Table 10, where we show TAda2D performs competitively in comparison with the models using the same

---

[3]The performances are referenced from the official release of the EK100 dataset (Damen et al., 2020).

Table 10: Comparison with the state-of-the-art approaches on Kinetics 400 (Kay et al., 2017).

| Model | Pretrain | Frames | GFLOPs | Top-1 | Top-5 |
|---|---|---|---|---|---|
| TSM (Lin et al., 2019a) | IN-1K | 8×3×10 | 43 | 74.1 | N/A |
| SmallBigNet (Li et al., 2020a) | IN-1K | 8×3×10 | 57 | 76.3 | 92.5 |
| TANet (Liu et al., 2021b) | IN-1K | 8×3×10 | 43 | 76.3 | 92.6 |
| TANet (Liu et al., 2021b) | IN-1K | 16×3×10 | 86 | 76.9 | 92.9 |
| SlowFast 4×16 (Feichtenhofer et al., 2019) | - | (4+32)×3×10 | 36.1 | 75.6 | 92.1 |
| SlowFast 8×8 (Feichtenhofer et al., 2019) | - | (8+32)×3×10 | 65.7 | 77.0 | 92.6 |
| TDN (Wang et al., 2021) | IN-1K | (8+32)×3×10 | 47 | 76.6 | 92.8 |
| TDN (Wang et al., 2021) | IN-1K | (16+64)×3×10 | 94 | 77.5 | 93.2 |
| CorrNet (Wang et al., 2020) | IN-1K | 32×1×10 | 115 | 77.2 | N/A |
| TAda2D (Ours) | IN-1K | 8×3×10 | 43 | 76.7 | 92.6 |
| TAda2D (Ours) | IN-1K | 16×3×10 | 86 | 77.4 | 93.1 |
| TAda2D$_{En}$ (Ours) | IN-1K | (8+16)×3×10 | 129 | 78.2 | 93.5 |
| MViT-B (Fan et al., 2021) | - | 16×1×5 | 70.5 | 78.4 | 93.5 |
| MViT-B (Fan et al., 2021) | - | 32×1×5 | 170 | 80.2 | 94.4 |
| TimeSformer (Bertasius et al., 2021) | IN-21K | 8×3×1 | 196 | 78.0 | 93.7 |
| ViViT-L (Arnab et al., 2021) | IN-21K | 16×3×4 | 1446 | 80.6 | 94.6 |
| Swin-T (Liu et al., 2021a) | IN-1K | 32×3×4 | 88 | 78.8 | 93.6 |
| TAdaConvNeXt-T (Ours) | IN-1K | 16×3×4 | 47 | 78.4 | 93.5 |
| TAdaConvNeXt-T (Ours) | IN-1K | 32×3×4 | 94 | 79.1 | 93.7 |

Table 11: Action localization evaluation on HACS and Epic-Kitchens-100. ↑ indicates the main evaluation metric for the dataset, *i.e.,* average mAP for action localization.

| Model | HACS | | | | | | Epic-Kitchens-100 | | | | | | |
|---|---|---|---|---|---|---|---|---|---|---|---|---|---|
| | @0.5 | @0.6 | @0.7 | @0.8 | @0.9 | **Avg.↑** | Task | @0.1 | @0.2 | @0.3 | @0.4 | @0.5 | **Avg.↑** |
| TSN | 43.6 | 37.7 | 31.9 | 24.6 | 15.0 | 28.6 | Verb | 15.98 | 15.01 | 14.09 | 12.25 | 10.01 | 13.47 |
| | | | | | | | Noun | 15.11 | 14.15 | 12.78 | 10.94 | 8.89 | 12.37 |
| | | | | | | | **Act.↑** | 10.24 | 9.61 | 8.94 | 7.96 | 6.79 | 8.71 |
| TAda2D | 48.7 | 42.7 | 36.2 | 28.1 | 17.3 | 32.3 | Verb | 19.70 | 18.49 | 17.41 | 15.50 | 12.78 | 16.78 |
| | | | | | | | Noun | 20.54 | 19.32 | 17.94 | 15.77 | 13.39 | 17.39 |
| | | | | | | | **Act.↑** | 15.15 | 14.32 | 13.59 | 12.18 | 10.65 | 13.18 |

backbone and the same number of frames. Compared with models with more frames, *e.g.,* TDN, TAda2D achieves a similar performance with less frames and computation. When compared to the more recent Transformer-based models, our TAdaConvNeXt-T provides competitive accuracy with similar or less computation.

# 6 EXPERIMENTS ON TEMPORAL ACTION LOCALIZATION

**Dataset, pipeline, and evaluation.** Action localization is an essential task for understanding untrimmed videos, whose current pipeline makes it heavily dependent on the quality of the video representations. We evaluate our TAda2D on two large-scale action localization datasets, HACS (Zhao et al., 2019) and Epic-Kitchens-100 (Damen et al., 2020). The general pipeline follows (Damen et al., 2020; Qing et al., 2021a;c). For evaluation, we follow the standard protocol for the respective dataset. We include the details on the training pipeline and the evaluation protocol in the Appendix C.

**Main results.** Table 11 shows that, compared to the baseline, TAda2D provides a stronger feature for temporal action localization, with an average improvement of over 4% on the average mAP on both datasets. In Appendix H, we further demonstrate the TAdaConv can also improve action localization when used as a plug-in module for existing models.

# 7 CONCLUSIONS

This work proposes Temproally-Adaptive Convolutions (TAdaConv) for video understanding, which dynamically calibrates the convolution weights for each frame based on its local and global temporal context in a video. TAdaConv shows superior temporal modelling abilities on both action classification and localization tasks, both as stand-alone and plug-in modules for existing models. We hope this work can facilitate further research in video understanding.

**Acknowledgement:** This research is supported by the Agency for Science, Technology and Research (A*STAR) under its AME Programmatic Funding Scheme (Project #A18A2b0046), by the RIE2020 Industry Alignment Fund – Industry Collaboration Projects (IAF-ICP) Funding Initiative, as well as cash and in-kind contribution from the industry partner(s), and by Alibaba Group through Alibaba Research Intern Program.

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

APPENDIX

In the appendix, we provide detailed analysis on the temporal convolutions (Appendix A), computational analysis (Appendix B), further implementation details (Appendix C) on the action classification and localization, model structures that we used for evaluation (Appendix D), per-category improvement analysis on Something-Something-V2 (Appendix E), further plug-in evaluations on Epic-Kitchens classification (Appendix G) plug-in evaluations on the temporal action localization task (Appendix H), the visualization of the training procedure of TSN and TAda2D (Appendix I), as well as detailed comparisons between TAdaConv and existing dynamic filters (Appendix J). Further, we show additional qualitative analysis in Fig. A3.

## A  DETAILED ANALYSIS ON TEMPORAL CONVOLUTIONS

Here, we provide detailed analysis to showcase the underlying process of temporal modelling by temporal convolutions. As in Sec. 3.1, we use depth-wise temporal convolutions for simplicity and its wide application. We first analyze the case where temporal convolutions are directly placed after spatial convolutions without non-linear activation in between, before activation functions is inserted in the second part of our analysis.

**Without activation.** We first consider a simple case with no non-linear activation functions between the temporal convolution and the spatial convolution. Given a $3 \times 1 \times 1$ depth-wise temporal convolution parameterized by $\boldsymbol{\beta} = [\boldsymbol{\beta}_1, \boldsymbol{\beta}_2, \boldsymbol{\beta}_3]$, where $\boldsymbol{\beta}_1, \boldsymbol{\beta}_2, \boldsymbol{\beta}_3 \in \mathbb{R}^{C_o}$, a spatial convolution parameterized by $\mathbf{W} \in \mathbb{R}^{C_o \times C_i \times k^2}$, the output feature $\tilde{\mathbf{x}}_t$ of the $t$-th frame can be obtained by:

$$\tilde{\mathbf{x}}_t = \boldsymbol{\beta}_1 \cdot (\mathbf{W} * \mathbf{x}_{t-1}) + \boldsymbol{\beta}_2 \cdot (\mathbf{W} * \mathbf{x}_t) + \boldsymbol{\beta}_3 \cdot (\mathbf{W} * \mathbf{x}_{t+1}) , \tag{8}$$

where $\cdot$ denotes element-wise multiplication with broadcasting, and $*$ denotes convolution over the spatial dimension. In this case, $\boldsymbol{\beta}$ could be grouped with the spatial convolution weight $\mathbf{W}$ and the combination of temporal and spatial convolution can be rewritten as Eq. 2 in the manuscript:

$$\tilde{\mathbf{x}}_t = \mathbf{W}_{t-1} * \mathbf{x}_{t-1} + \mathbf{W}_t * \mathbf{x}_t + \mathbf{W}_{t+1} * \mathbf{x}_{t+1} , \tag{2}$$

where $\mathbf{W}_{t-1} = \boldsymbol{\beta}_1 \cdot \mathbf{W}$, $\mathbf{W}_t = \boldsymbol{\beta}_2 \cdot \mathbf{W}$ and $\mathbf{W}_{t+1} = \boldsymbol{\beta}_3 \cdot \mathbf{W}$. This equation share the same form with the Eq. 2 in the manucript. In this case, the combination of temporal convolution with spatial convolution can be certainly viewed as the temporal convolution is simply performing calibration on spatial convolutions before aggregation, with different weights assigned to different time steps for the calibration.

**With activation.** Next, we consider a case where activation is in between the temporal convolution and spatial convolution. The output feature $\tilde{\mathbf{x}}_t$ are now obtained by Eq. 1 in the manuscript:

$$\tilde{\mathbf{x}}_t = \boldsymbol{\beta}_1 \cdot \delta(\mathbf{W} * \mathbf{x}_{t-1}) + \boldsymbol{\beta}_2 \cdot \delta(\mathbf{W} * \mathbf{x}_t) + \boldsymbol{\beta}_3 \cdot \delta(\mathbf{W} * \mathbf{x}_{t+1}) . \tag{1}$$

Next, we show that this can be still rewritten in the form of Eq. 2. Here, we consider the case where ReLU (Nair & Hinton, 2010) is used as the activation function, denoted as $\delta$:

$$\delta(x) = \begin{cases} x & x > 0 \\ 0 & x \leq 0 \end{cases} . \tag{9}$$

Hence, the term $\delta(\mathbf{W} * \mathbf{x}_t)$ can be easily expressed as:

$$\delta(\mathbf{W} * \mathbf{x}_t) = \mathbf{M}_t \cdot \mathbf{W} * \mathbf{x}_t , \tag{10}$$

where $\mathbf{M}_t \in \mathbb{R}^{C \times H \times W}$ is a binary map sharing the same shape as $\mathbf{x}_t$, indicating whether the corresponding element in $\mathbf{W} * \mathbf{x}_t$ is greater than 0 or not. That is:

$$\mathbf{M}_t^{(c,i,j)} = \begin{cases} 1 & \text{if} \quad (\mathbf{W} * \mathbf{x}_t)^{(c,i,j)} > 0 \\ 0 & \text{if} \quad (\mathbf{W} * \mathbf{x}_t)^{(c,i,j)} \leq 0 \end{cases} , \tag{11}$$

where $c, i, j$ are the location index in the tensor. Hence, Eq. 1 can be expressed as:

$$\tilde{\mathbf{x}}_t = \boldsymbol{\beta}_1 \cdot \mathbf{M}_{t-1} \cdot \mathbf{W} * \mathbf{x}_{t-1} + \boldsymbol{\beta}_2 \cdot \mathbf{M}_t \cdot \mathbf{W} * \mathbf{x}_t + \boldsymbol{\beta}_3 \cdot \mathbf{M}_{t+1} \cdot \mathbf{W} * \mathbf{x}_{t+1} . \tag{1}$$

In this case, we can set $\mathbf{W}_{t-1}^{(i,j)} = \boldsymbol{\beta}_1 \cdot \mathbf{M}_{t-1}^{(i,j)} \cdot \mathbf{W}$, $\mathbf{W}_t^{(i,j)} = \boldsymbol{\beta}_2 \cdot \mathbf{M}_t^{(i,j)} \cdot \mathbf{W}$, and $\mathbf{W}_{t+1}^{(i,j)} = \boldsymbol{\beta}_3 \cdot \mathbf{M}_{t+1}^{(i,j)} \cdot \mathbf{W}$, where $(i, j)$ indicate the spatial location index. In this case, each filter for a specific time step $t$ is composed of $H \times W$ filters and Eq. 1 can be rewritten as Eq. 2. Interestingly, it can be observed that with ReLU activation function, the convolution weights are different for all spatio-temporal locations, since the binary map $\mathbf{M}$ depends on the results of the spatial convolutions.

Table A1: Comparison of different operations for spatial and temporal modeling (Lin et al., 2019a; Tran et al., 2018; Wang et al., 2020; Hara et al., 2018). 'T.' refers to the temporal modeling ability.

| T. | Operation | Parameters | FLOPs |
|---|---|---|---|
| ✗ | Spat. conv | $C_o \times C_i \times k^2$ | $C_o \times C_i \times k^2 \times THW$ |
| ✓ | Temp. conv | $C_o \times C_i \times k$ | $C_o \times C_i \times k \times THW$ |
| ✓ | Temp. shift | $C_o \times C_i \times k^2$ | $C_o \times C_i \times k^2 \times THW$ |
| ✓ | (2+1)D conv | $C_o \times C_i \times (k^2 + k)$ | $C_o \times C_i \times (k^2 + k) \times THW$ |
| ✓ | 3D conv | $C_o \times C_i \times k^3$ | $C_o \times C_i \times k^3 \times THW$ |
| ✓ | Correlation | $C_i \times T \times k^2$ | $C_i \times k^2 \times THW$ |
| ✓ | TAdaConv | $C_o \times C_i \times k^2 + 2 \times C_i \times C_i/r \times k$ | $C_o \times C_i \times k^2 \times THW + C_i \times (THW + T)$ $+C_i \times C_i/r \times (2 \times k \times T + 1) + C_o \times C_i \times k^2 \times T$ |

## B  COMPUTATIONAL ANALYSIS

Consider the input tensor with the shape of $C_i \times T \times H \times W$, where $C_i$ denotes the number of input channels, TAdaConv essentially performs 2D convolution, with weights dynamically generated. Hence, a good proportion of the computation is carried out by the 2D convolutions:

$$\text{FLOPs(Conv2D)} = C_o \times C_i \times k^2 \times T \times H \times W$$
$$\text{Params(Conv2D)} = C_o \times C_i \times k^2$$

where $C_o$ denote the number of output channels, and $k$ denotes the kernel size of the 2D convolutions. For the weight generation, the features are first aggregated by average pooling over the spatial dimension and the temporal dimension, which contains no parameters:

$$\text{FLOPs(GAP}_{\text{spatial}}) = C_i \times T \times H \times W$$
$$\text{FLOPs(GAP}_{\text{temporal}}) = C_i \times T$$

For the local information, a two layer 1D convolution with kernel size of $k_w$ are applied, with reduction ratio of $r$ in between. For global information, a one layer 1D convolution with kernel size of 1 is applied.

$$\text{FLOPs(Gen)} = 2 \times C_i \times C_i/r \times k \times T + C_i \times C_i/r$$
$$\text{Params(Gen)} = 2 \times C_i \times C_i/r \times k + C_i \times C_i/r$$

Further, the calibration weight $\boldsymbol{\alpha} \in \mathbb{R}^{C_i \times T}$ is multiplied to the kernel weight of 2D convolutions $\mathbf{W} \in \mathbb{R}^{C_o \times C_i \times k^2}$:

$$\text{FLOPs(Calibration)} = C_o \times C_i \times k^2 \times T ,$$

Hence, the overall computation and parameters are:

$$
\begin{aligned}
\text{FLOPs(TAdaConv)} =& \text{FLOPs(Conv2D)} + \text{FLOPs(GAP)} + \text{FLOPs(Gen)} + \text{FLOPs(Calibration)} \\
=& C_o \times C_i \times k^2 \times THW + C_i \times THW + C_i \times T \\
& + 2 \times C_i \times C_i/r \times k \times T + C_i \times C_i/r + C_o \times C_i \times k^2 \times T
\end{aligned}
$$

$$
\begin{aligned}
\text{Params(TAdaConv)} =& \text{Params(Conv2D)} + \text{Params(Gen)} \\
=& C_o \times C_i \times k^2 + 2 \times C_i \times C_i/r \times k + C_i \times C_i/r
\end{aligned}
$$

For the FLOPs, despite the overwhelming number of terms, all the other terms are at least an order of magnitude smaller than the first term. This is in contrast to the (2+1)D convolutions for spatio-temporal modelling, where the FLOPs are $C_o \times C_i \times (k^2 + k) \times THW$. The extra computation introduced by the temporal convolutions is only $k$ times smaller than the 2D convolutions. Table A1 shows the comparison of computation and parameters with other approaches.

## C  FURTHER IMPLEMENTATION DETAILS

Here, we further describe the implementation details for the action classification and action localization experiments. For fair comparison, we keep all the training strategies the same for our baseline, the plug-in evaluations as well as our own models.

## C.1 ACTION CLASSIFICATION

Our experiments on the action classification are conducted on three large-scale datasets. For all action classification models, we train them with synchronized SGD using 16 GPUs. The batch size for each GPU is 16 and 8 respectively for 8-frame and 16-frame models. The weights in TAda2D are initialized using ImageNet (Deng et al., 2009) pre-trained weights (He et al., 2016), except for the calibration function $\mathcal{G}$ and the batchnorm statistics (BN$_2$) in the average pooling branch. In the calibration function, we randomly initialize the first convolution layer (for non-linear weight generation) following He et al. (2015), and fill zero to the weight of last convolution layer. The batchnorm statistics are initialized to be zero so that the initial state behaves the same as without the average pooling branch. For all models, we use a dropout ratio (Hinton et al., 2012) of 0.5 before the classification heads. Spatially, we randomly resize the short side of the video to [256, 320] and crop a region of 224×224 to the network in ablation studies, and set the scale to [224, 340] following TANet (Liu et al., 2021b) for comparison against the state-of-the-arts. Temporally, we perform interval based sampling for Kinetics-400 and Epic-Kitchens-100, with interval of 8 for 8 frames, interval of 5 for 16 frames and interval of 2 for 32 frames. On Something-Something-V2, we perform segment based sampling.

On *Kinetics-400*, a half-period cosine schedule is applied for decaying the learning rate following Feichtenhofer et al. (2019), with the base learning rate set to 0.24 for ResNet-base models using SGD. For TAdaConvNeXt, the base learning rate is set to 0.0001 for the backbone and 0.001 for the head, using adamw (Loshchilov & Hutter, 2017) as the optimizer. The models are trained for 100 epochs. In the first 8 epochs, we adopt a linear warm-up strategy starting from a learning rate of 0.01. The weight decay is set to 1e-4. The frames are sampled based on a fixed interval, which is 8 for 8-frame models, 5 for 16-frame models and 2 for 32-frame models. Additionally for TAdaConvNeXt-T, a drop-path rate of 0.4 is employed.

On *Epic-Kitchens-100*, the models are initialized with weights pre-trained on Kinetics-400, and are further fine-tuned following a similar strategy as in kinetics. The training length is reduced to 50 epochs, with 10 epochs for warm-up. The base learning rate is 0.48. Following Damen et al. (2020) and Huang et al. (2021), we connect two separate heads for predicting verbs and nouns. Action predictions are obtained according to the strategies in Huang et al. (2021), which is shown to have a higher accuracy over the original one in Damen et al. (2020). For fair comparison, we also trained and evaluated our baseline using the same strategy.

On *Something-Something-V2*, we initialize the model with ImageNet pretrained weights for ResNet-based models and Kinetics-400 pre-trained weights for TAdaConvNeXt. A segment-based sampling strategy is adopted, where for $T$-frame models, the video is divided into $T$ segments before one frame is sampled from each segment randomly for training or uniformly for evaluation. The models are trained for 64 epochs, with the first 4 being the warm-up epochs. The base learning rate is set to 0.48 in training TAda2D with SGD, and 0.001/0.0001 respectively for the head and the backbone in training TAdaConvNeXt with adamw. Following Liu et al. (2021a), we use stronger augmentations such as mixup (Zhang et al., 2017), cutmix (Yun et al., 2019), random erasing (Zhong et al., 2020) and randaugment (Cubuk et al., 2020) with the same parameters in Liu et al. (2022).

It is worth noting that, for SlowFast models (Feichtenhofer et al., 2019) in the plug-in evaluations, we do not apply precise batch normalization statistics in our implementation as in its open-sourced codes, which is possibly the reason why our re-implemented performance is slightly lower than the original published numbers.

## C.2 ACTION LOCALIZATION

We evaluate our model on the action localization task using two large-scale datasets. The overall pipeline for our action localization evaluation is divided into finetuning the classification models, obtaining action proposals and classifying the proposals.

**Finetuning.** On *Epic-Kitchens*, we simply use the evaluated action classification model. On *HACS*, following (Qing et al., 2021c), we initialize the model with Kinetics-400 pre-trained weights and train the model with adamW for 30 epochs (8 warmups) using 32 GPUs. The mini-batch size is 16 videos per GPU. The base learning rate is set to 0.0002, with cosine learning rate decay as in Kinetics. In our case, only the segments with action labels are used for training.

Table A2: Model structure of R3D, R(2+1)D and R2D that we used in our experiments. **Blue** and **green** fonts indicate respectively the default convolution operation and optional operation that can be replaced by TAdaConv. (Better viewed in color.)

| Stage | R3D | R(2+1)D | R2D (default baseline) | output sizes |
|---|---|---|---|---|
| Sampling | interval 8, $1^2$ | interval 8, $1^2$ | interval 8, $1^2$ | $8\times224\times224$ |
| $conv_1$ | $3\times7^2$, 64 
 stride 1, $2^2$ | $1\times7^2$, 64 
 stride 1, $2^2$ | $1\times7^2$, 64 
 stride 1, $2^2$ | $8\times112\times112$ |
| $res_2$ | $\begin{bmatrix} 1\times1^2, 64 \\ \mathbf{3\times3^2, 64} \\ 1\times1^2, 256 \end{bmatrix}\times3$ | $\begin{bmatrix} 1\times1^2, 64 \\ \mathbf{1\times3^2, 64} \\ \mathbf{3\times1^2, 64} \\ 1\times1^2, 256 \end{bmatrix}\times3$ | $\begin{bmatrix} 1\times1^2, 64 \\ \mathbf{1\times3^2, 64} \\ 1\times1^2, 256 \end{bmatrix}\times3$ | $8\times56\times56$ |
| $res_3$ | $\begin{bmatrix} 1\times1^2, 128 \\ \mathbf{3\times3^2, 128} \\ 1\times1^2, 512 \end{bmatrix}\times4$ | $\begin{bmatrix} 1\times1^2, 128 \\ \mathbf{1\times3^2, 128} \\ \mathbf{3\times1^2, 128} \\ 1\times1^2, 512 \end{bmatrix}\times4$ | $\begin{bmatrix} 1\times1^2, 128 \\ \mathbf{1\times3^2, 128} \\ 1\times1^2, 512 \end{bmatrix}\times4$ | $8\times28\times28$ |
| $res_4$ | $\begin{bmatrix} 1\times1^2, 256 \\ \mathbf{3\times3^2, 256} \\ 1\times1^2, 1024 \end{bmatrix}\times6$ | $\begin{bmatrix} 1\times1^2, 256 \\ \mathbf{1\times3^2, 256} \\ \mathbf{3\times1^2, 256} \\ 1\times1^2, 1024 \end{bmatrix}\times6$ | $\begin{bmatrix} 1\times1^2, 256 \\ \mathbf{1\times3^2, 256} \\ 1\times1^2, 1024 \end{bmatrix}\times6$ | $8\times14\times14$ |
| $res_5$ | $\begin{bmatrix} 1\times1^2, 512 \\ \mathbf{3\times3^2, 512} \\ 1\times1^2, 2048 \end{bmatrix}\times3$ | $\begin{bmatrix} 1\times1^2, 512 \\ \mathbf{1\times3^2, 512} \\ \mathbf{3\times1^2, 512} \\ 1\times1^2, 2048 \end{bmatrix}\times3$ | $\begin{bmatrix} 1\times1^2, 512 \\ \mathbf{1\times3^2, 512} \\ 1\times1^2, 2048 \end{bmatrix}\times3$ | $8\times7\times7$ |
| global average pool, fc | | | | $1\times1\times1$ |

**Proposal generation.** For the action proposals, a boundary matching network (BMN) (Lin et al., 2019b) is trained over the extracted features on the two datasets. On *Epic-Kitchens*, we extract features with the videos uniformly decoded at 60 FPS. For each clip, we use 8 frames with an interval of 8 to be consistent with finetuning, which means a feature roughly covers a video clip of one seconds. The interval between each clip for feature extraction is 8 frames (*i.e.,* 0.133 sec) as well. The shorter side of the video is resized to 224 and we feed the whole spatial region into the backbone to retain as much information as possible. Following Qing et al. (2021a), we generate proposals using BMN based on sliding windows. The predictions on the overlapped region of different sliding windows are simply averaged. On *HACS*, the videos are decoded at 30 FPS, and extend the interval between clips to be 16 (*i.e.,* 0.533 sec) because the actions in HACS last much longer than in Epic-Kitchens. The shorter side is resized to 128 for efficient processing. For the settings in generating proposals, we mainly follow Qing et al. (2021c), except that the temporal resolution is resized to 100 in our case instead of 200.

**Classification.** On *Epic-Kitchens*, we classify the proposals with the fine-tuned model using 6 clips. Spatially, to comply with the feature extraction process, we resize the shorter side to 224 and feed the whole spatial region to the model for classification. On *HACS*, considering the property of the dataset that only one action category can exist in a video, we obtain the video level classification results by classifying the video level features, following Qing et al. (2021c).

**Evaluation.** For evaluation, we follow the standard evaluation protocol used in the respective datasets, *i.e.,* the average mean Average Precision (average mAP) at IoU threshold [0.5:0.05:0.95] for HACS (Zhao et al., 2019) and [0.1:0.1:0.5] for Epic-Kitchens-100 (Damen et al., 2020).

## D  MODEL STRUCTURES

The detailed model structures for R2D, R(2+1)D and R3D is specified in Table A2. We highlight the convolutions that are replaced by TAdaConv by default or optionally. For all of our models, a small modification is made in that we remove the max pooling layer after the first convolution and set the spatial stride of the second stage to be 2, following Wang et al. (2020). Temporal resolution is kept unchanged following recent works (Feichtenhofer et al., 2019; Li et al., 2020b; Jiang et al., 2019). Our *R3D* is obtained by simply expanding the R2D baseline in the temporal dimension by a factor

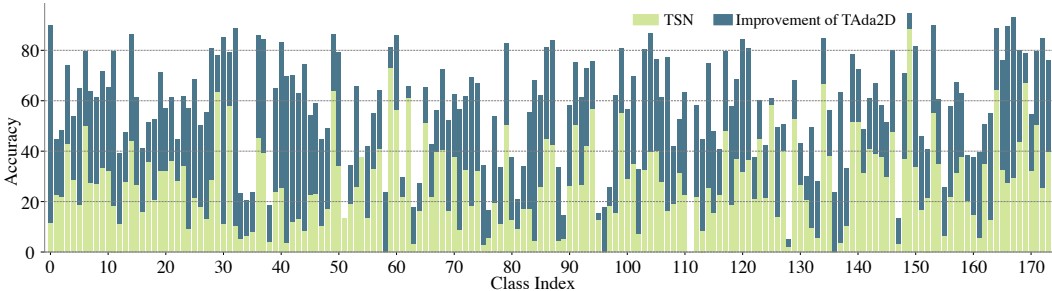

Figure A1: **Per-category performance comparison of TAda2D against the baseline TSN.** We achieve an average per-category performance improvement of 30.35%.

Table A3: Comparison with the state-of-the-art approaches over action classification on Epic-Kitchens-100 (Damen et al., 2020). ↑ indicates the main evaluation metric for the dataset. For fair comparison, we implement all the baseline models using our own training strategies.

| Model | Frames | GFLOPs | Params. | Top-1 | | | Top-5 | | |
|---|---|---|---|---|---|---|---|---|---|
| | | | | Act.↑ | Verb | Noun | Act.↑ | Verb | Noun |
| SlowFast 4×16 | 4+32 | 36.10 | 34.5M | 38.17 | 63.54 | 48.79 | 58.68 | 89.75 | 73.37 |
| SlowFast 4×16 + TAdaConv | 4+32 | 36.11 | 37.7M | 39.14 | 64.50 | 49.59 | 59.21 | 89.67 | 73.88 |
| SlowFast 8×8 | 8+32 | 65.71 | 34.5M | 40.08 | 65.05 | 50.72 | 60.10 | 90.04 | 74.26 |
| SlowFast 8×8 + TAdaConv | 8+32 | 65.73 | 37.7M | 41.35 | 66.36 | 52.32 | 61.68 | 90.59 | 75.89 |
| R(2+1)D | 8 | 49.55 | 28.1M | 37.45 | 62.92 | 48.27 | 58.02 | 89.75 | 73.60 |
| R(2+1)D + TAdaConv$_{2d}$ | 8 | 49.57 | 31.3M | 39.72 | 64.48 | 50.26 | 60.22 | 90.01 | 75.06 |
| R(2+1)D + TAdaConv$_{2d+1d}$ | 8 | 49.58 | 34.4M | 40.10 | 64.77 | 50.28 | 60.45 | 89.99 | 75.55 |
| R3D | 8 | 84.23 | 47.0M | 36.67 | 61.92 | 47.87 | 57.47 | 89.02 | 73.05 |
| R3D + TAdaConv$_{3d}$ | 8 | 84.24 | 50.1M | 39.30 | 64.03 | 49.94 | 59.67 | 89.84 | 74.56 |

of three. We initialize with weights reduced by 3 times, which means the original weight is evenly distributed in adjacent time steps. We construct the *R(2+1)D* by adding a temporal convolution operation after the spatial convolution. The temporal convolution can also be optionally replaced by TAdaConv, as shown in Table 3 and Table A3. For its initialization, the temporal convolution weights are randomly initialized, while the others are initialized with the pre-trained weights on ImageNet. For SlowFast models, we keep all the model structures identical to the original work (Feichtenhofer et al., 2019).

For TAdaConvNeXt, we keep most of the model architectures as in ConvNeXt (Liu et al., 2022), except that we use a tubelet embedding similar to (Arnab et al., 2021), with a size of 3×4×4 and stride of 2×4×4. Center initialization is used as in (Arnab et al., 2021). Based on this, we simply replace the depth-wise convolutions with TAdaConv to construct TAdaConvNeXt.

## E  PER-CATEGORY IMPROVEMENT ANALYSIS ON SSV2

This section provides a per-category improvement analysis on the Something-Something-V2 dataset in Fig.A1. As shown in Table 5, our TAda2D achieves an overall improvement of 31.7%. Our per-category analysis shows an mean improvement of 30.35% over all the classes. The largest improvement is observed in class 0 (78.5%, *Approaching something with your camera*), 32 (78.4%, *Moving away from something with your camera*), 30 (74.3%, *Lifting up one end of something without letting it drop down*), 44 (66.2%, *Moving something towards the camera*) and 41 (66.1%, *Moving something away from the camera*). Most of these categories contain large movements across the whole video, whose improvement benefits from temporal reasoning over the global spatial context. For class 30, most of its actions lasts a long time (as it needs to be determined whether the end of something is let down or not). The improvements over the baseline mostly benefits from the global temporal context that are included in the weight generation process.

Table A4: Ablation studies.

(a) Ablation studies on kernel size with linear calibration weight generation function.

| Kernel size | Top-1 |
|---|---|
| 1 | 37.5 |
| 3 | 56.5 |
| 5 | 57.3 |
| 7 | 56.5 |

(b) Ablation studies on kernel size with non-linear calibration weight generation function.

| | $K_2$=1 | $K_2$=3 | $K_2$=5 | $K_2$=7 |
|---|---|---|---|---|
| $K_1$=1 | 36.8 | 57.1 | 57.8 | 57.9 |
| $K_1$=3 | 57.3 | 57.8 | 57.9 | 58.0 |
| $K_1$=5 | 57.6 | 57.9 | 58.2 | 57.9 |
| $K_1$=7 | 57.4 | 57.6 | 58.0 | 57.6 |

(c) Ablation studies on reduction ratio $r$ for $K_1 = K_2 = 3$.

| Ratio $r$ | Top-1 |
|---|---|
| 1 | 57.79 |
| 2 | 57.83 |
| 4 | 57.78 |
| 8 | 57.66 |

Table A5: Plug-in evaluation of TAdaConv on the action localization on HACS and Epic-Kitchens. ↑ indicates the main evaluation metric for the dataset. 'S.F.' is SlowFast network.

| Model | HACS | | | | | | Epic-Kitchen-100 | | | | | | |
|---|---|---|---|---|---|---|---|---|---|---|---|---|---|
| | @0.5 | @0.6 | @0.7 | @0.8 | @0.9 | **Avg.↑** | Task | @0.1 | @0.2 | @0.3 | @0.4 | @0.5 | **Avg.↑** |
| S.F. 8×8 | 50.0 | 44.1 | 37.7 | 29.6 | 18.4 | 33.7 | Verb | 19.93 | 18.92 | 17.90 | 16.08 | 13.24 | 17.21 |
| | | | | | | | Noun | 17.93 | 16.83 | 15.53 | 13.68 | 11.41 | 15.07 |
| | | | | | | | **Act.↑** | 14.00 | 13.19 | 12.37 | 11.18 | 9.52 | 12.04 |
| S.F. 8×8 + TAdaConv | 51.7 | 45.7 | 39.3 | 31.0 | 19.5 | 35.1 | Verb | 19.96 | 18.71 | 17.65 | 15.41 | 13.35 | 17.01 |
| | | | | | | | Noun | 20.17 | 18.90 | 17.58 | 15.83 | 13.18 | 17.13 |
| | | | | | | | **Act.↑** | 14.90 | 14.12 | 13.32 | 12.07 | 10.57 | 13.00 |

# F    FURTHER ABLATION STUDIES

Here we provide further ablation studies on the kernel size in the calibration weight generation. As shown in Table A4a and Table A4b, kernel size does not affect the classification much, as long as the temporal context is considered. Further, Table A4c shows the sensitivity analysis on the reduction ratio, which demonstrate the robustness of our approach against different set of hyper-parameters.

# G    FURTHER PLUG-IN EVALUATION FOR TADACONV ON CLASSIFICATION

In complement to Table 3, we further show in Table A3 the plug-in evaluation on the action classification task on the Epic-Kitchens-100 dataset. As in the plug-in evaluation on Kinetics and Something-Something-V2, we compare performances with and without TAdaConv over three baseline models, SlowFast (Feichtenhofer et al., 2019), R(2+1)D (Tran et al., 2018) and R3D (Hara et al., 2018) respectively representing three kinds of temporal modeling techniques. The results are in line with our observation in Table 3. Over all three kinds of temporal modelling strategies, adding TAdaConv further improves the recognition accuracy of the model.

# H    PLUG-IN EVALUATION FOR TADACONV ON ACTION LOCALIZATION

Here, we show the plug-in evaluation on the temporal action localization task. Specifically, we use SlowFast as our baseline, as it is shown to be superior in the localization performance in Qing et al. (2021b) compared to many early backbones. The result is presented in Table A5. With TAdaConv, the average mAP on HACS is improved by 1.4%, and the average mAP on Epic-Kitchens-100 action localization is improved by 1.0%.

# I    COMPARISON OF TRAINING PROCEDURE

In this section, we compare the training procedure of TSN and TAda2D on Kinetics-400 and Something-Something-V2. The results are presented in Fig. A2. TAda2D demonstrates a stronger fitting ability and generality even from the early stages of the training, despite that the initial state of TAda2D is identical to that of TSN.

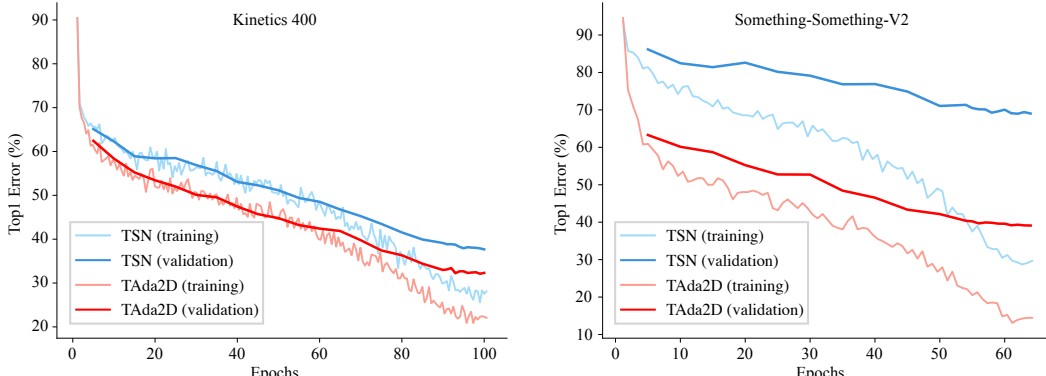

Figure A2: **Training and validation on Kinetics-400 and Something-Something-V2.** On both datasets, TAda2D shows a stronger capability of fitting the data and a better generality to the validation set. Further, TAda2D reduces the overfitting problem in Something-Something-V2.

Table A6: Approach comparison between different dynamic filters. The weights column denotes how weights in respective approaches are obtained. The pre-trained weights colmun shows whether the weight generation can exploit pre-trained models such as ResNet (He et al., 2016).

| Operations | Weights | Temporal Modelling | Location Adaptive | Pretrained weights |
|---|---|:---:|:---:|:---:|
| CondConv | Mixture of experts $\mathbf{W} = \sum_n f(\mathbf{x})_n \mathbf{W}_n$ | ✗ | ✗ | ✗ |
| DynamicFilter | Completely generated $\mathbf{W} = g(\mathbf{x})$ | ✗ | ✗ | ✗ |
| DDF | Completely generated $\mathbf{W} = g(\mathbf{x})$ | ✗ | ✓ | ✗ |
| TAM | Completely generated $\mathbf{W} = g(\mathbf{x})$ | ✓ | ✗ | ✗ |
| TAdaConv | Calibrated from a base weight $\mathbf{W} = h(\mathbf{x})\mathbf{W}_b$ | ✓ | ✓ | ✓ |

## J    COMPARISON WITH EXISTING DYNAMIC FILTERS

In this section, we compare our TAdaConv with previous dynamic filters in two perspectives, respectively the difference in the methodology and in the performance.

### J.1    COMPARISON IN THE METHODOLOGY LEVEL

For the former comparison, we include Table A6 to show the differences in different approaches, which is a full version of Table 1. We compare TAdaConv with several representative approaches in image and in videos, respectively CondConv (Yang et al., 2019), DynamicFilter (Jia et al., 2016), DDF (Zhou et al., 2021) and TAM (Liu et al., 2021b).

The first difference in the methodology level lies in the source of weights, where previous approaches obtain weights by mixture of experts or generation completely dependent on the input. *Mixture of experts* denotes $\mathbf{W} = \sum_n \alpha_n \mathbf{W}_n$, where $\alpha_n$ is a scalar obtained by a function $f$, *i.e.,* $\mathbf{W} = \sum_n f(\mathbf{x})_n \mathbf{W}_n$. *Completely generated* means the weights are only dependent on the input, *i.e.,* $\mathbf{W} = g((\mathbf{x}))$, where $g$ generates complete kernel for the convolution. In comparison, the weights in TAdaConv are obtained by *calibration*, *i.e,,* $\mathbf{W} = \alpha \mathbf{W}_b$, where $\alpha$ is a vector calibration weight and $\alpha = h((\mathbf{x}))$ where $h(.)$ generates the calibration vector for the convolutions. Hence, this fundamental difference in how to obtain the convolution weights makes the previous approaches difficult to exploit pre-trained weights, while TAdaConv can easily load pre-trained weights in $\mathbf{W}_b$. This ability is essential for video models to speed up the convergence.

The second difference lies in the ability to perform temporal modelling. The ability to perform temporal modelling does not only mean the ability to generate weights according to the whole sequence in dynamic filters for videos, but it also requires the model to generate different weights for the same set of frames with different orders. For example, weights generated by the global descriptor obtained by global average pooling over the whole video $\text{GAP}_{st}$ does not have the temporal modelling ability, since they can not generate different weights if the order of the frames in the input sequence are reversed or randomized. Hence, most image based approaches based on global de-

Table A7: Performance comparison with other dynamic filters. *Our Init.* denotes initializing the calibration weights to ones so that the initial calibrated weights is identical to the pre-trained weights. Temp. Varying is short for temporally varying, which indicates different weights for different temporal locations (frames). * denotes that the branch was originally not designed for generating filter or calibration weights, but we slightly modified the structure so that it can be used for calibration weight generation. **(Numbers in brackets)** show the performance improvement brought by our initialization scheme for calibration weights.

| Calibration Generation | Our Init. | Temp. Varying | Generation source | Top-1 |
|---|---|---|---|---|
| DynamicFilter | ✗ | ✗ | $\text{GAP}_{st}(\mathbf{x})(C \times 1)$ | 41.7 |
| DDF-like | ✗ | ✓ | $\text{GAP}_{st}(\mathbf{x})(C \times 1)$ | 49.8 |
| TAM (global branch) | ✗ | ✗ | $\text{GAP}_{s}(\mathbf{x})(C \times T)$ | 39.7 |
| TAM (local*+global branch) | ✗ | ✓ | $\text{GAP}_{s}(\mathbf{x})(C \times T)$ | 41.3 |
| DynamicFilter | ✓ | ✗ | $\text{GAP}_{st}(\mathbf{x})(C \times 1)$ | 51.2 (+9.5) |
| DDF-like | ✓ | ✓ | $\text{GAP}_{st}(\mathbf{x})(C \times 1)$ | 53.8 (+4.0) |
| TAM (global branch) | ✓ | ✗ | $\text{GAP}_{s}(\mathbf{x})(C \times T)$ | 52.9 (+13.2) |
| TAM (local*+global branch) | ✓ | ✓ | $\text{GAP}_{s}(\mathbf{x})(C \times T)$ | 54.3 (+13.0) |
| TAdaConv w/o global info $\mathbf{g}$ | ✓ | ✓ | $\text{GAP}_{s}(\mathbf{x})(C \times T)$ | 57.9 |
| TAdaConv | ✓ | ✓ | both $\text{GAP}_{st}(\mathbf{x})(C \times 1)$ and $\text{GAP}_{s}(\mathbf{x})(C \times T)$ | 59.2 |

scriptor vectors (such as CondConv and DynamicFilter) or based on adjacent spatial contents (DDF) can not achieve temporal modelling. TAM generates convolution weights for temporal convolutions based on temporally local descriptors obtained by the global average pooling over the spatial dimension $\text{GAP}_s$, which yields different weights if the sequence changes. Hence, in this sense, TAM has the temporal modelling abilities. In contrast, TAdaConv exploits both temporally local and global descriptors to utilize not only local but also global temporal contexts. Details on the source of the weight generation process is also shown in Table A7.

The third difference lies in whether the weights generatd are shared for different locations. For CondConv, DynamicFilter and TAM, their generated weights are shared for all locations, while for DDF, the weights are varied according to spatial locations. In comparison, TAdaConv generate temporally adaptive weights.

## J.2    COMPARISON IN THE PERFORMANCE LEVEL

Since TAdaConv is fundamentally different from previous approaches in the generation of calibration weights, it is difficult to directly compare the performance on video modelling, especially for those that are not designed for video modelling. However, since the calibration weight in TAdaConv $\boldsymbol{\alpha}$ is completely generated, *i.e.*, $\boldsymbol{\alpha} = f((\mathbf{x}))$, we can use other dynamic filters to generate the calibration weights for TAdaConv. Since MoE based approaches such as CondConv were essentially designed for applications with less memory constraint but high computation requirements, it is not suitable for video applications since it would be too memory-heavy for video models. Hence, we apply approaches that generate complete kernel weights to generate calibration weights, and compare them with TAdaConv. The performance is listed in Table A7.

It is worth noting that these approaches originally generate weights that are randomly initialized. However, as is shown in Table 6, our initialization strategy for the calibration weights are essential for yielding reasonable results, we further apply our initialization on these existing approaches to see whether their generation function is better than the one in TAdaConv. In the following paragraphs, we provide details for applying representative previous dynamic filters in TAdaConv to generate the calibration weight.

For DynamicFilter (Jia et al., 2016), the calibration weight $\boldsymbol{\alpha}$ is generated using an MLP over the global descriptor that is obtained by performing global average pooling over the whole input $\text{GAP}_{st}$, *i.e.*, $\boldsymbol{\alpha} = \text{MLP}(\text{GAP}_{st}(\mathbf{x}))$. In this case, the calibration weights are shared between different time steps.

For DDF (Zhou et al., 2021), we only use the channel branch since it is shown in Table 7 that it is better to leave the spatial structure unchanged for the base kernel. Similarly, the weights in DDF are also generated by applying an MLP over the global descriptor, *i.e.*, $\boldsymbol{\alpha} = \text{MLP}(\text{GAP}_{st}(\mathbf{x}))$. The

difference between DDF and DynamicFilter is that for different time step, DDF generates a different calibration weight.

The original structure of TAM (Liu et al., 2021b) only generates kernel weights with its global branch, and uses local branch to generate attention maps over different time steps. In our experiments, we modify the TAM a little bit and further make the local branch to generate kernel calibration weights as well. Hence, for only-global version of TAM, the calibration weights are calculated as follows: $\boldsymbol{\alpha} = \mathcal{G}(\mathrm{GAP}_s(\mathbf{x}))$, where $\mathrm{GAP}_s$ denotes global average pooling over the spatial dimension and $\mathcal{G}$ denotes the global branch in TAM. In this case, calibration weights are shared for all temporal locations. For local+global version of TAM, the calibration weight are calculated by combining the results of the local $\mathcal{L}$ and the global branch $\mathcal{G}$, *i.e.,* $\boldsymbol{\alpha} = \mathcal{G}(\mathrm{GAP}_s(\mathbf{x})) \cdot \mathcal{L}(\mathrm{GAP}_s(\mathbf{x}))$, where $\cdot$ denotes element-wise multiplication with broadcasting. This means in this case, the calibration weights are temporally adaptive. Note that this is our modified version of TAM. The original TAM does not have a temporally adaptive convolution weights.

The results in Table A7 show that (a) without our initialization strategy, previous approaches that generate random weights at initialization are not suitable for generating the calibration weights in TAdaConv; (b) our initialization strategy can conveniently change this and make previous approaches yield reasonable performance when they are used for generating calibration weights; and (c) the calibration weight generation function in TAdaConv, which combines the local and global context, outperform all previous approaches for calibration.

Further, when we compare TAdaConv without global information with TAM (local*+global branch), it can be seen that although both approach generates temporally varying weights from the frame descriptors $\mathrm{GAP}_s(\mathbf{x})$ with shape $C \times T$, our TAdaConv achieves a notably higer performance. Adding the global information enables TAdaConv to achieve a more notable lead in the comparison with previous dynamic filters.

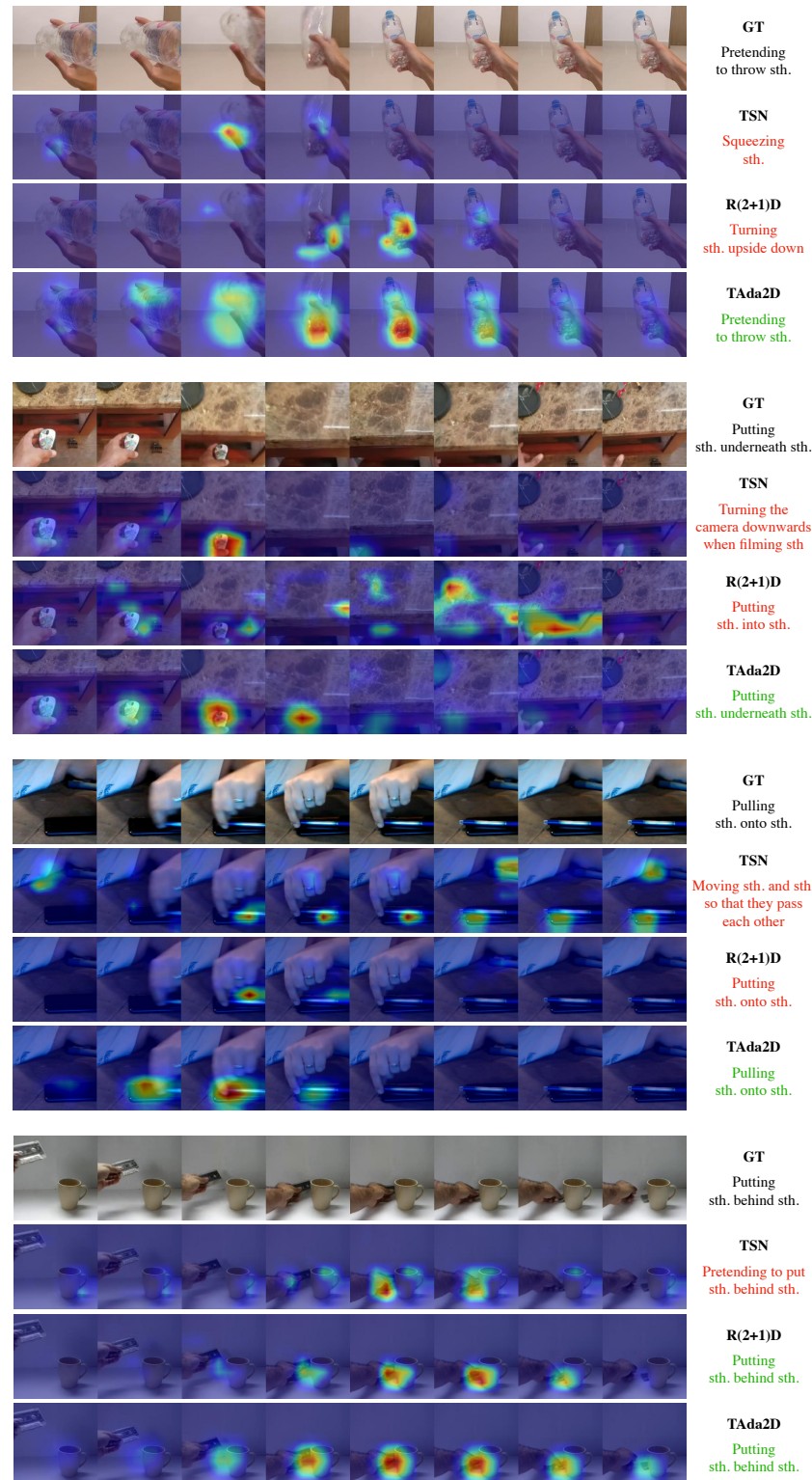

Figure A3: **Further qualitative evaluations on the Something-Something-V2 dataset.** In most cases, TAda2D captures meaningful areas in the videos for the correct classification. Further, the activated region of TAda2D also lasts longer along the temporal dimension compared to other two models, thanks to the global temproal context in the weight generation function $\mathcal{G}$.

