# OpenReview forum: "TAda! Temporally-Adaptive Convolutions for Video Understanding"
_ICLR.cc/2022/Conference — ICLR 2022 Poster_

### Official Review · Reviewer_jnMz · 2021-10-29

**Correctness:** 3
**Technical Novelty And Significance:** 2
**Empirical Novelty And Significance:** 2
**Recommendation:** 5
**Confidence:** 4

**Main Review:**

**Strengths**
- The paper is well written and easy to follow and understand. The approach is well described.
- The idea of generating weights is interesting and has not been greatly studied in video recognition.
- The paper conducts experiments on multiple standard datasets, so it can be compared to other approaches.


**Weaknesses**

- Eq. 1 and 2 are a bit misleading. The addition of the activation function (and normalization layers) leads to a non-linearity in the network, which has been shown to be the critical part of NNs. Omitting this part is leaving out a key piece of NNs. It makes it seem like (2+1)D convolution is just a multiplication of the weights, when in fact the non-linearity is important. This should be corrected to clarify the connection and differences between the proposed approach and convolutions.

- Similarity to previous works. The approach is not especially novel, the idea of generating temporal weights from 2d weights has been shown before. For example, see "3D Inflated TGM layer" in "Evolving Space-Time Neural Architectures for Videos", ICCV'19, as well as all the (2+1)D CNN works (e.g., S3D, " A Closer Look at Spatiotemporal Convolutions for Action Recognition",CVPR'18, etc.). It seems the main difference is that the temporal weights are dynamically generated here. However, there are no experiments showing the difference between static, learned weights and dynamic weights. Adding this experiment would strengthen the paper.

- Table 1 is quite hard to understand and see the difference between the approaches. It would be helpful to have a table comparing say a ResNet-50 with each of those configurations (or a variety of networks) to show how the flops and parameters varies. As is, it appears that this approach is less than 3D conv, but more that the others? (This seems to be part of table 2, but table 1 still is quite hard to follow).

- The claims of state-of-the-art performance are wrong. Many results on Kinetics-400 are omitted (see https://paperswithcode.com/sota/action-classification-on-kinetics-400 have many over 82% while this paper reports a best of 77.9). The claims and/or table should be updated and corrected/clarified. Similarly, S-Sv2 (https://paperswithcode.com/sota/action-recognition-in-videos-on-something) has numbers over the best reported 66.7% in this paper and epic kitchens (https://paperswithcode.com/sota/action-recognition-on-epic-kitchens-100) well over the 40.89% reported.



**Summary Of The Paper:**

This paper proposes a modified version of convolution for videos where a 2D conv kernel is converted to a 3D conv kernel by multiplying the 2d weights by a per-frame value. This creates a set of weights for 3D convolution that are dynamic for each frame. The approach is evaluated on multiple video datasets, and shows some benefit over their baseline.

**Summary Of The Review:**

Overall, the paper makes an incremental contribution by essentially making (2+1)D convolution adaptive by generating the 1D conv weights within the network. The experimental claims need clarifications and revisions, as they are currently omitting stronger performing models. The ablation experiments could be strengthened by including experiments on the effect of dynamic vs. static weights. Overall the paper makes an incremental contribution and the experiments are not currently strong enough to show the value of it. With revisions, the paper could be much stronger.

---

> ### Author Response · Authors · 2021-11-22
> **Response to Reviewer jnMz (3/3)**
>
> 4. **Clearer FLOPS and parameters comparison in Table 1.**
>
> Thanks for the suggestion. For Table 1 (now the *Table 2* in the latest manuscript), we have added direct comparison with temporal convolutions, which includes example values when they are added to spatial convolutions and the standard ResNet. This shows a clear superiority of the efficiency of TAdaConv, in that temporal convolutions introduces additional 33% and 15% computation to spatial convolutions and ResNet, while TAdaConv adds additional 0.2% and 0.06%.
>
> ---
>
> 5. **Claim of the state-of-the-art performance.**
>
> Thanks for pointing this out.
>
> - Yes, it is true that we do not report and compare with the models that report the best accuracy on these datasets. This is because many of those models require a large amount of computation due to the size of the model as well as the spatio-temporal resolution of the input sequence, which both have a huge influence on the final performance. Because of the hardware constraint, we are unable to match the model size or the resolution of those models. Hence, for fair comparison, TAda2D is compared against convolutional models with the same depth and similar inputs. We achieve the state-of-the-art performances among models that are designed and trained at a similar scale. This follows the standard pracise in convolutional video models such as TSM (ICCV 2019) [1], bLVNet (NeurIPS 2019)[2], SmallBig (CVPR 2020) [3], TDN (CVPR 2021) [4], and TANet (ICCV 2021) [5], *etc.*
> - Since TAdaConv is designed as a plug-in module to enhance any type of convolutions in existing models (which is demonstrated to be effective for many models and for all 1D, 2D and 3D convolutions in *Table 3* of the latest manuscript), TAdaConv can also be used to enhance those models that reports the best accuracy if they use convolutions.
> - Further, as reviewer *6Gv3* pointed out, TAdaConv can also be included in the search space in neural architecture search for action recognition, which can also be helpful to the comminity to potentially obtain stronger video models.
>
> ----
> **References**
>
> [1] Lin, Ji, Chuang Gan, and Song Han. "Tsm: Temporal shift module for efficient video understanding." In ICCV 2019.
>
> [2] Fan, Quanfu, et al. "More is less: Learning efficient video representations by big-little network and depthwise temporal aggregation." In NeurIPS 2019.
>
> [3] Li, Xianhang, et al. "Smallbignet: Integrating core and contextual views for video classification." In CVPR 2020.
>
> [4] Wang, Limin, et al. "TDN: Temporal difference networks for efficient action recognition." In CVPR 2021.
>
> [5] Liu, Zhaoyang, et al. "Tam: Temporal adaptive module for video recognition." In ICCV 2021.

---

> ### Author Response · Authors · 2021-11-22
> **Response to Reviewer jnMz (2/3)**
>
> 3. **Similarity to previous works.**
>
> Our apologies for the possible disorientation caused. Here, we explain the essential nature of TAdaConv first, before we compare TAdaConv with the "3D inflated TGM layer" and "(2+1)D".
> - Essential nature of TAdaConv:
>     - The essential meaning of "temporally adaptive" in TAdaConv is that the convolution weights are not shared for different time steps. Hence, converting a 2D conv into 2D TAdaConv means making the 2D conv that originally shares weights for all frames to have different weights for different time steps, where the different weights for different frames are generated from a base 2D conv kernel, as in *Figure 1* in the manuscript. This means a 2D TAdaConv still performs 2D convolutions.
>     - Essentially, converting a 2D conv into a 2D TAdaConv is "converting a 2D conv kernel into *a set of 2D conv kernels*". This is fundamentally different from "converting a 2D conv kernel into *a 3D conv kernel*", in that
>       - *Operation:*
> a set of 2D conv kernels still performs 2D convolutions, while a 3D conv kernel performs 3D convolution.
>       - *Invariance:*
> a set of 2D conv kernels have different weights for different time steps varying from frame to frame, while a 3D conv kernel shares weights for different time steps.
>     - Indeed, we can also have 3D TAdaConv or 1D TAdaConv by converting the 3D conv kernels or 1D conv kernels into a set of 3D conv kernels or 1D conv kernels. But still, the essential operation is not changed during this convertion, *i.e.,* a n-D conv converted to TAdaConv is still a n-D conv with different weights on different time steps.
> - Difference of TAdaConv from the "3D Inflated TGM layer":
>     - *Operation:*
>       - The "3D Inflated TGM layer" inflates a 2D conv kernel into a 3D conv kernel. The keyword is "inflated": it obtains a set of 2D conv kernels and combine them into a 3D conv kernel. This means it essentially is a 3D convolution operation, with $L$-times more computation than a 2D convolution operation. Hence, it also has around $L$-times more computation than 2D TAdaConv.
>       - In contrast, a 2D TAdaConv converts a 2D conv kernel into a set of 2D conv kernels, without combining them together to form a 3D conv kernel. The keyword for TAdaConv is "temporally adaptive": we make a convolution that originally shares weights for different time steps to have different weights on different frames. After convertion, a 2D TAdaConv is still a 2D conv, with $L$-times less computation than a 3D convolution operation.
>       - Indeed, we can have 3D TAdaConv, but the temporally adaptive 3D conv kernels are obtained from a base 3d conv kernel. There is no "inflation" in TAdaConv.
>     - *Invariance:*
>       - The key of "3D Inflated TGM layer" lies in how it obtains the 3D conv weights, which means it does not change the 3D convolution itself. Hence, it has the property of spatio-temporal invariance, which means shared weights for all spatiotemporal locations.
>       - TAdaConv essentially relaxes the invariance along the temporal dimension, by applying different weights on different time steps.
>     - *Compatibility:*
>
>         Further, as we show in the plug-in evaluation in *Table 3 and Table A3* (the original Table 2 and Table A2), TAdaConv can achieve notable improvement by making the 3d conv to be temporally adaptive, *i.e.,* to have different weights for spatiotemporal aggregation in different time steps. Since "3D Inflated TGM layer" obtains a 3D conv kernel and performs standard 3D convolution that shares weights for all frames, it can be further enhanced by TAdaConv.
> - Difference of TAdaConv from the "(2+1)D":
>     - *Operation:*
>       - R(2+1)D achieves temporal modelling by attaching the standard 1D convolution to the 2D convolutions. This, as we show in *Table 2* in the latest version of our manuscript, induces non-negligible computation (33% increase in the operation level and 15% increase in the network level if applied to standard ResNet).
>       - Based on 2D convolutions, TAdaConv aims to directly empower the spatial convolutions to have temporal modelling abilities, by calibrating the convolution kernel for each frame dynamicall according to its temporal context. This gives us improved efficiency. As shown in *Table 2*, TAdaConv introduces only 0.2% computation increase (in the operation level) to spatial convolutions and 0.06% (in the network level) to ResNet, which is significantly smaller than R(2+1)D.
>     - *Invariance:*
>       - Both 2D and 1D convolutions in R(2+1)D uses standard convolutions, which shares weights for all frames.
>       - Both 2D and 1D TAdaConv dynamically ajusts the convolution weight for each frame.
>     - *Compatibility:*
>
>     Further, as we show in the plug-in evaluation in *Table 3 and Table A3* (the original Table 2 and Table A2), TAdaConv brings notable improvement to R(2+1)D by making the 2D and 1D convolutions in R(2+1)D temporally adaptive.

---

> ### Author Response · Authors · 2021-11-22
> **Response to Reviewer jnMz (1/3)**
>
> Thank you for your acknowledgement of our strengths. We address your concerns as follows:
>
> ---
>
> 1. **Eq. 1 and 2 are a bit misleading.**
>
> Our intention of including Eq.1 and 2 was to show the general idea that when temporal convolution is connected to spatial convolutions, it essentailly calibrates the spatial convolutions, before temporal aggregation. This has served as an inspiration for us to directly learn to calibrate the spatial convolution weights for different temporal locations.
>
> Indeed, as you pointed out, non-linearity is an essential part of neural networks. Hence, we added non-linearity into the formulation. In *Appendix A*, we show that with ReLU between spatial and temporal convolutions, temporal convolution still calibrates the spatial convolutions, except that it now calibrates the spatial convolution with different calibration weight for all spatio-temporal locations.
> Nevertheless, this does not affect that we take inspiration from the calibration process, and propose our TAdaConv.
>
> ----
>
> 2. **Experiments showing the difference between static, learned weights and dynamic weights.**
>
> Thanks for the valuable suggestion. Using static, learned calibration weights to generate different convolution weights for different frames indeed helps us to validate that: the relaxation of temporal invariance can benefit temporal modelling.
> We compare different ways of generating the calibration weights in *Table R6* (which is also included in the *Table 6* of the latest manuscript).
> The details of the experiments are as follows:
>
> - *Baseline:*
>
>     The baseline is TSN with no calibration at all. It has no temporal reasoning abilities and hence has limited accuracy.
>
> - *Learnable calibration:*
>   - Details:
>
>     Learnable calibration introduces a learnable weight to calibrate the 2D convolution kernels. In this case, if it is temporally varying, the shape of learnable weights is $T\times C$ with $T$ and $C$ being the number of frames and channels respectively. If it is temporal invariant, it uses a $1\times C$ learnable calibration weight for all frames.
>   - Results:
>
>     Since temporally varying learnable calibration significantly outperforms temporal invariant one, it validates our hypothesis that relaxed temporal invariance benefits temporal modelling.
>  - *Dynamic calibration:*
>    - Details:
>
>         For the dynamic calibration here, we use a simple global description vector obtained by global average pooling over the input to generate the calibration weights. The weights are generated with a two-layer MLP. For temporally varying dynamic calibration, it maps the global description vector $1\times C$ to $1\times TC$ and reshape to $T\times C$ for different frames. For temporally invariant calibration, the global description vector is mapped to $1\times C$ for all frames.
>    - Results:
>
>         Dynamic calibration outperforms temporally varying learnable calibration even without relaxing the temporal invaraince. Introducing temporally varying calibration to dynamic further raises the performance. This further validates our hypothesis.
>  - Further, using our generation function to generate temporally varying weights yields the best performance among these variants.
>
> **Table R6** Comparison between different forms of calibration weights (*Table 6* in the latest manuscript).
>
> | Calibration weight | Temporally Varying | Top-1 |
> | ------------------ | ------------------ | ----- |
> | None (TSN)         | No                 | 32.0  |
> | Learnable          | No                 | 34.3  |
> | Learnable          | Yes                | 45.4  |
> | Dynamic            | No                 | 51.2  |
> | Dynamic            | Yes                | 53.4  |
> | TAdaConv           | Yes                | 59.2  |

---

> ### Author Response · Authors · 2021-12-01
> **Sincerely expecting further feedbacks**
>
> Dear Reviewer jnMz,
>
> We would like to thank you again for your time and effort in reviewing our manuscript. We have carefully revised our submission according to your insightful comments, and put in our best efforts to clarify the questions and concerns. It would be greatly appreciated if you could check our responses and provide your valuable feedback. We would be more than glad to provide further clarifications.
>
> Thank you and sincerely looking forward to your further comments.
>
> Best regards,
>
> Authors

---

### Official Review · Reviewer_4biu · 2021-11-02

**Correctness:** 3
**Technical Novelty And Significance:** 2
**Empirical Novelty And Significance:** 3
**Recommendation:** 6
**Confidence:** 4

**Main Review:**

I like the general idea of adaptive weight calibration and its simplicity. The paper was easy to read and understand, and the experimental results are extensive and show good performance. I also appreciate the detailed information presented in the supplementary material.
The following are some concerns that I would like to point out.

1) There are no comparisons with any of the dynamic modules. The idea of content-adaptive weights (or modules) has been extensively explored in the area of dynamic networks. However, this paper lacks experiments that compare the performance of TAda2D and previous approaches. In its current form, the paper only shows that using TAda2D can bring some performance gain when compared to TSN (Table 3(a) and Figure 3). I believe that more experiments should be performed to justify the use of TAda2D. For example, is TAda2D the most effective way to calibrate the convolutioanl weights? I do not think that I could find the answer from the current version of the paper.

2) I have another concern about the scalability of TAda2D. In Tables 5 and 7, using more frames brings only a small performance gain for TAda2D. For example, 16fx2x3 increases Top-1 score by 4.3% for TSM but 1.4$ for TAda2D. I am wondering if it implies that TAda2D works better on shorter input video clips.

3) It looks like TAda2D and TANet are very similar in terms of their adaptive characteristics across the temporal dimension. I believe that TAda2D should be compared more deeply with TANet. It is also interesting that TANet and TAda2D have similar patterns in Table 7 (76.3% with 43x30 GFLOPS and 76.9% with 86x30 GFLOPS).

4) I think it would be better to add a column for GFLOPS and #Params for Table 5 and A2 because the paper put weight on TAdaConv's efficiency. In addition, it is a bit hard to grasp Table 1. It would be better to add example values as already done in footnote 2.

**Summary Of The Paper:**

This paper proposes TAdaConv that calibrates kernel weights of convolutional layers adaptively according to the temporal dynamics of the input tensor. It is designed to incorporate both the local and global temporal context by using stacked two-layer 1D convolutional operations and global average pooling. Since it works exactly the same as the original convolutional layer at the initial stage, it is easy to insert TAdaConv into existing ConvNet architectures. On top of this, the authors construct TAda2D networks by introducing the temporal feature aggregation module that is based on a strided temporal average pooling. Many experiments performed on video/action classification and localization demonstrate the effectiveness of the proposed module.

**Summary Of The Review:**

The paper is well-motivated and easy to understand. There are a few things I would like to point out: 1) Comparison with dynamic modules and TANet, 2) Scalability for longer input clips, 3) Clearly show the efficiency of TAda2D.

---

> ### Author Response · Authors · 2021-11-22
> **Response to Reviewer 4biu (2/2)**
>
> 3. **Comparison with TANet.**
>
> We would first like to thank you for this insightful observation that our TAda2D had the same performance with TANet on Kinetics-400. Inspired by this, we take a deep look into their implementation and found the spatial augmentation of TANet is slightly stronger than TAda2D. Hence, when we aligned the augmentations, we obtain a higher performance (for 8f: ours 76.7 vs. TANet 76.3 and for 16f: ours 77.4 vs. TANet 76.9). This also raises the performance of TAda2D on the other benchmarks, which we have all updated into the latest version of the manuscript.
>
> Back to the question itself, yes, our TAda2D is to some extent similar to TANet since we both belong to the dynamic filter family, but they are different in the following major ways:
> - *Operations:*
>   - TAM/TANet proposes adaptive weights for the *temporal convolutions*
>   - TAdaConv can be used to enhance *any convolutions* in videos (*Table 3* in the manuscript)
> - *Temporal invariance:*
>   - TAM/TANet generates weights that are shared for different frames.
>   - TAdaConv calibrates weights so that they are different for different frames.
> - *How the weights are obtained:*
>   - TAM/TANet obtains weights by directly generating from the input, *i.e.,* $\mathbf{W}=g(x)$.
>   - TAdaConv obtains weights by calibrating on a base weight, *i.e.,* $\mathbf{W}=h(x)\mathbf{W}_b$.
> - *Source for generation of weights:*
>   - As shown in the *Table R5* above, TAM/TANet generates weights solely over frame-level description vectors
>   - TAdaConv generates weights with consideration of both frame-level and global description vectors.
>
> Further, when we apply TAM for generating calibration weights, it yields suboptimal performance without our initialization scheme for the calibration weight generation. With our initialization scheme, it underperforms TAdaConv, even when we modify the local branch for generating temporally adaptive weights (shown in *Table A7* in *Appendix J.2*).
>
> ----
>
> 4. **GFLOPS and #Params for Table 5 and A2.**
>
> Thank you for the suggestion.
>
> For Table 5 (now *Table 8* in the latest manuscript), we showed a visualization plot of *Performance vs. GFLOPs* in Figure 3 (b), demonstrating the superior performance-accuracy trade-off of TAda2D. For *Table A2* (now *Table A3* in the latest appendix), we have added the GFLOPS and number of parameters. These results show that TAdaConv improves the model performance with negligible additional computation.
>
> ----
>
> 5. **Inclusion of example values in Table 1.**
>
> Thank you for the suggestion. For a clearer demonstration,
>    - we have replaced the original Table 1 with a comparison between (2+1)D convolution and TAdaConv with *Table 2* in the latest manuscript, which includes
>      - notation based computation and paramter analysis;
>      - operation level example value;
>      - network level example value.
>
>      This more straitforwardly show the computation superiority of TAdaConv compared to (2+1)D convolutions, with our additional computations over the spatial convolution and ResNet is only 6% (0.002/0.308) and 0.4% (0.02/4.94) of the additional computation introduced by temporal convolutions.
>    - we moved the notation based computation and paramter comparision with other temporal modelling approaches to *Appendix B*, where we detailedly perform computational analysis on our TAdaConv.
>
> We hope these would make everything clear now.
>
> ----
> **References**
>
> [1] Yang, Brandon, et al. "CondConv: conditionally parameterized convolutions for efficient inference." In NeurIPS 2019.
>
> [2] Jia, Xu, et al. "Dynamic filter networks." In NeurIPS 2016.
>
> [3] Zhou, Jingkai, et al. "Decoupled Dynamic Filter Networks." In CVPR 2021.
>
> [4] Liu, Zhaoyang, et al. "Tam: Temporal adaptive module for video recognition." In ICCV 2021.

---

> > ### Comment · Reviewer_4biu · 2021-11-30
> > **I appreciate the rebuttal**
> >
> > I am glad that my review helped you discover the alignment of the augmentation scheme brings some additional gain. I appreciate the thorough rebuttal. It addresses all my concerns, so I will adjust my rating.

---

> > > ### Author Response · Authors · 2021-11-30
> > > **Many thanks for the acknowledgement of our latest manuscript**
> > >
> > > Thank you again for the insightful suggestions that helped improve our manuscript. We appreciate your adjustment of the final rating.

---

> ### Author Response · Authors · 2021-11-22
> **Response to Reviewer 4biu (1/2)**
>
> Thank you for your appreciation of our approach, our writing and our experiments. We appreciate your comments and address your concerns as follows:
>
> ----
>  1. **Comparison with dynamic modules.**
>
> Thanks for the valuable suggestion. We compare our TAdaConv with previous dynamic filter based approaches in the following two aspects. We have also included more detailed explanation in the latest version of our paper in *Appendix J*:
>
> - *Approach comparison* (included in *Sec 3.2* and *Appendix J.1*)
>
>     Indeed, content adaptive weights are extensively explored in dynamic filtering, which generally falls into the following categories: (a) mixture of experts (MoE) based (such as CondConv [1]), (b) completely generated weights (such as DynamicFilter [2] and DDF[3] in image as well as TAM[4] in video). The second can be divided into location adaptive ones with different weights for different locations (DDF) and location invariant with shared weights for all locations (TAM and DynamicFilter). The difference between them are listed in the *Table R4* here. This comparison is also included in the revised paper in *Table 1* and more detailedly in *Table A6* in *Appendix J*. In conclusion, from the approach level, TAdaConv can simultaneously
>     - perform temporal modelling
>     - be temporally adaptive
>     - effectively exploit pretrained weights
> - *Empirical comparison* (included in *Appendix J.2*)
>
>     Since most previous image-based approaches did not consider temporal modelling, directly replacing TAdaConv with them to compare performances would be unfair. However, we can apply those approaches in TAdaConv to generate the calibration weights. We only compare with approaches that directly generate weights, since MoE based approaches are too memory-heavy for video models. The results are shown in the *Table A7* in the latest manuscript (as well as *Table R5* here). The table shows that
>     - when used for generating calibration weights, previous approaches can hardly exploit pre-trained weights because the generated calibration weights of these approaches are random at initialization
>     - our initialization scheme effectively lead previous dynamic filters to yield reasonable performances (over 50%) when used as a calibration weight
>     - the calibration weights generated by TAdaConv are the most effective ones compared to previous approaches
>
> **Table R4** approach comparison between dynamic filters and TAdaConv (also in *Table A6* in *Appendix J*). Notations: $f$ generates scalar weight, $g$ generates the weights for the complete kernel and $h$ generates vector calibration weights for the base weight $\mathbf{W}_b$.
>
>    | Operation | Weights | Temporal Modelling | Location Adaptive | Pre-trained weights |
>    | --------- | ---- | ------------------ | ----------------- | ------------------- |
>    | CondConv [1]  | MoE $\mathbf{W}=\sum_nf(x)_n\mathbf{W}_n$ | No | No | No |
>    | DynamicFilter [2] | completely generated $\mathbf{W}=g(x)$ | No | No | No |
>    | DDF [3] | completely generated $\mathbf{W}=g(x)$ | No | Yes | No |
>    | TAM [4] | completely generated $\mathbf{W}=g(x)$ |Yes | No | No |
>    | TAdaConv (Ours) | Calibrated from base $\mathbf{W}=h(x)\mathbf{W}_b$ | Yes | Yes | Yes |
>
> **Table R5** empirical comparison on SSV2 between dynamic filters and TAdaConv for calibration generation (also more detailedly in *Table A7* in the Appendix).
>
> | Calibration generation function | Temporally Varying | Source of generation | Top 1 Acc w/o our init |  Top 1 Acc w/ our init|
> | --------- | ------------------ | --------------------- | --------- | --------- |
> | Dynamic Filter | No | $\text{GAP}_{st}(\mathbf{x})$ (Cx1) | 41.7 | 51.2 |
> | TAM (global branch)| No | $\text{GAP}_{s}(\mathbf{x})$ (CxT) | 39.7 | 52.9 |
> | DDF (channel) | Yes | $\text{GAP}_{st}(\mathbf{x})$ (Cx1) | 49.8 | 53.8 |
> | TAdaConv | Yes | $\text{GAP}_{st}(\mathbf{x})$ and $\text{GAP}_s(\mathbf{x})$ (Cx1 & CxT)| 47.8 | 59.2 |
>
> ----
> 2. **Scalability of TAda2D.**
>
> Thanks for the comments. Scalability is indeed an import aspect to consider. In the latest version of our manuscript, we included a visualization of *Performance vs. GFLOPs* on Something-Something-V2 in *Figure 3*. It shows that
> - The scalability of TSM is indeed exceptional, compared to existing models.
> - Our model is as scalable as other state-of-the-arts models except for TSM.

---

### Official Review · Reviewer_DgYX · 2021-11-02

**Correctness:** 4
**Technical Novelty And Significance:** 3
**Empirical Novelty And Significance:** 3
**Recommendation:** 8
**Confidence:** 3

**Main Review:**

Strengths:
- The main strength of the paper lies in the novelty of applying calibration weights directly to the spatial weights in the temporal dimension. Significant previous works have explored various approaches where excitation is applied to the output features in the temporal dimension, but this approach appears to have benefits in terms of computation.
- The set of experiments is well laid out and there are sufficient ablation studies over the TAdaConv components. Each significant design decision appears to be tested in the experiments.
- As a drop-in replacement for spatial convolution, it shows clear performance improvements for established architectures. It is clear that this dynamic temporal adaption is something which can be beneficial broadly in convolution-based video-understanding models.

Weaknesses:
- Computational comparison - it would have been useful to have a visualisation plot of performance vs GFLOPS to be able to compare the model against other state of the art models, given that this is one of the main benefits of the model.
- The introduction to the paper is not precise on the motivation for the work. The 3rd paragraph introduces what is sought, ("integrating the dynamic weight calibration into the spatial convolutions"), but not why it is sought. It would be useful for the reader to understand why this would be a better approach, and what are the deficiencies of the existing approaches that necessitate this.
- Performance comparisons to TDN are prefaced with the fact that TDN uses more frames. I am not sure if this is fair, given the GFLOPs of this model are in line despite the larger number of frames according to Table 7.
- The language could do with a little bit of finessing in the paper. There are some instances of grammar mistakes and some unwieldy sentences that should be made more clear to improve readability. For examples the last sentences in the 'Calibration dimension' and 'Different proportion of channels employing TAdaConv' paras on page 7.
- Some care could be made to clarify what is meant by a term before its use. For example, linear vs nonlinear weight generation, temporally sensitive on page 3.

**Summary Of The Paper:**

This work seeks to improve performance of video understanding models though the use of spatial convolutions which are dynamically adapted in time when applied to a sequence of frames. The novelty of this work lies in the fact that the weights of the spatial convolutions are adapted, rather than applying a temporal excitation to feature maps to achieve the same purpose. To achieve this, a drop-in replacement for spatial convolutions is designed which uses pooled local and global frame descriptors to produce the temporal calibration weights. Furthermore, this operation is used as part of a temporal feature aggregation block that is used as part of 2D CNN architecture. The benefits of the approach include that it can make effective use of pre-trained weights and that it can improve model capacity for a marginal increase in computation. Experimentally, it is show large performance improvements over a baseline and match state of the art on significant datasets.

**Summary Of The Review:**

I believe that this work presents a novel idea of weight calibration that has benefits in terms of performance and model efficiency that for video understanding tasks. Its use as a drop-in replacement shows that it improves performance for existing architectures. It also shows that there may still be unexplored aspects for improving these model types. Despite the issues that I see in the paper, which I believe could be straightforwardly improved, I think this paper is at an acceptable threshold.

---

> ### Author Response · Authors · 2021-11-22
> **Response to Reviewer DgYX (1/2)**
>
> Thank you for your acknowledgement of our novelty, the thoroughness of the experiments and the potential influences. We also appreciate your valuable comments and address your concerns as follows:
>
> ----
>
> 1. **Visualization plot of performance vs. GFLOPs.**
>
> Thanks, we have included the visualization in Figure 3, which shows the superiority of TAda2D against recent state-of-the-art approaches in terms of performance/computation tradeoff.
>
> ----
>
> 2. **Precise motivation for TAdaConv.**
>
> Thanks for the suggestion. As we pointed out in our updated manuscript and the general response above, the motivation of "integrating the dynamic weight calibration into spatial convolutions" is considered as the following three aspects:
>
>    - *Why spatial convolutions?*
>
>      Since the prevalently used temporal convolutions still introduce non-negligible computation over spatial convolutions, we set out to directly let spatial convolutions to have temporal modelling abilities.
>
>    - *Why temporally adaptive?*
>
>      As networks with spatially varying weights (such as DDF [1] and LRLC [2]) have demonstrated the superiority of the relaxed spatial invariance in modelling complex spatial contents, we hypothesis that relaxing the invariance along the temporal dimension can also benefit temporal modelling. This hypothesis is validated in *Table 6* in the latest manuscript, where temporally varying calibrations (both learnable and dynamic) outperform temporally shared ones. Further, making the convolutions in existing video models in *Table 3* and *Table A3* in the latest manuscript also show the benefit of making convolutions temporally adaptive.
>
>    - *Why calibration?*
>
>      Because of the deficiency of previous dynamic filter models(a), we take inspiration from the calibration nature of temporal convolutions(b) and design calibration on the spatial convolution kernels.
>      - (a) Deficiency: Most existing dynamic filters can not exploit pre-trained weights such as that of ResNet. In fact, previous works did not have to achieve this, since most of them are trained on imagenet. However, pre-trained networks are essential for video mdoels to reduce the training duration. For instance, SlowFast [3] without pre-trainining on ImageNet is trained for 196 or 256 epochs on Kinetics-400, while most video models using pre-trained weights only requires around 100 epochs, such as TSM [4], TDN [5] and TANet [6], *etc*. By the calibration on a base weight, TAdaConv can load the pre-trained weights to the base weight.
>      - (b) Temporal convolution essentially performs calibration on the spatial convolutions, where it is applied in most works with spatial convolutions loaded with pretrained weights, such as TANet [6] and STM [7], *etc*. We show in *Sec 3.1* as well as more detailedly in *Appendix A* that temporal convolutions conntected to spatial convolutions (with and without ReLU in between) perform calibration to the spatial convolution weights.
>
> ----
>
> 3. **Deficiency of previous works that necessitate TAdaConv.**
>
> As in the latest version of *Sec 1 Para 3*, deficiency of previous dynamic filters are:
>
> - pre-traiend weights can hardly be exploited, since their weights are completely generated from the input;
> - temporal reasoning can not be well performed,
>   - for image models: since their weights are generated either with respect to the spatial content or the global content;
>   - for video models (TAM): since it generates 1D convolution from temporally local descriptors (C$\times$T) and *shares* the convolution weight for all frames, which can not benefit from temporally varying weights.
>
> In contrast, TAdaConv can
> - leverage pre-trained weights since the weights in TAdaConv is generated by a learnable base weight and a content-adaptive calibration weight;
> - perform temporal reasoning better,
>   - since the weights are generated from both global and local temporal context for each frame (validated in *Table R3* down below);
>   - since TAdaConv benefits from relaxed temporal invariance which brings improved temporal modelling abilities (validated in *Table 6* in the latest manuscript)
>
> ***Table R3*** Comparison of different source of calibration for generating temporally adaptive calibration weights.
>
>    | Source for calibration weight generation | global (C$\times$1) | local (C$\times$T) | global + local (C$\times$1 + C$\times$T)|
>    | - | ------ | ----- | -------------- |
>    | Top1 |  53.8 (*Table 6*)  |  57.8 (*Table 4*) | 59.2 (*Table 4*) |

---

> > ### Comment · Reviewer_DgYX · 2021-11-29
> > **Response to latest revisions**
> >
> > Thanks to the authors for providing comprehensive details of all the revisions made to the manuscript. I believe that much of the concerns of my original review have been well addressed. In particular:
> >
> > 1. This addition is very helpful to the reader and certainly improves the manuscript.
> > 2. The revisions have made these points much clearer, especially in Sec 1.
> > 3. The revised Sec 1 as well as the Tables A6 and A7 in response to reviewer 4biu have strengthened the claims of a deficiency in the research.
> > 4. On this point, I still have slight misgivings, mostly due to how fairness is being defined. If it is based on computation, then I believe it is fair to compare with TDN. If it is on number of frames used, then it would be unfair to compare. As the claims of the paper mostly relate to computation, I would argue that this the most applicable criterion, especially as the strength of the TDN work is its ability to access information from a large number of frames in a small computational budget. Perhaps, quantifying computation differently (e.g. by including frame decoding cost) could make this more concrete. Besides this point, it is reassuring to note that the latest performance is competitive with TDN.
> > 5. 6. Agreed, the latest manuscript is improved in this regards.
> >
> > My initial review score was based on the assumption that these revisions could be straightforwardly addressed and the level of technical significance. However, given the improvements in clarity and claims of the paper, I will adjust my rating.

---

> > > ### Author Response · Authors · 2021-11-30
> > > **Many thanks for the acknowledgement of our latest manuscript**
> > >
> > > Thank you for the time and effort in the detailed reply to our responses.
> > >
> > > For the comparison to TDN, we totally agree that the measurement of fairness plays an important role in deciding whether we should compare our performance to TDN. Indeed, including frame decoding cost is a reasonable way when comparing the efficiency of different models. We will try to include this in the later version of our manuscript.

---

> ### Author Response · Authors · 2021-11-22
> **Response to Reviewer DgYX (2/2)**
>
> 4. **Performance comparison to TDN.**
>
>    - Indeed, the GFLOPS of TDN is only slightly higher than TAda2D (*TDN: 36x1.306=~47GFLOPs* vs. *Ours: 43GFLOPs*). It is also true that the S-TDM of TDN takes input from more frames (as illustrated by Figure 2 and Figure 3(a) in TDN [5], 4 additional frames for each frame), which means TDN receives more information from the video. To some extent, this makes the comparison between our models unfair.
>    - Additionally, we aligned the augmentation of TAda2D with TANet [6] and found TAda2D performs competitively against TDN (*Table 8* and *Table 10*).
>    - Further, since TDN is also a convolutional model, which means TAdaConv can also be employed in replacement of the 2D convolutions in TDN for further improvement.
>
> ----
>
> 5. **Language.**
>
> Thanks, we have gone through the paper and modified the ones that we found were expressed in a too complicated way, including what you have mentioned.
>
> ----
>
> 6. **Explanation before term.**
>
> Thanks for the suggestion. We have replaced "temporally sensitive" with "temporally adaptive" to be consistent with everywhere else in the paper, and explained the linear and non-linear weight generation before the description of the results. We have also gone through the paper and added all the explanations that we found was missing.
>
> ----
>
> **References**
>
>  [1] Zhou, Jingkai, et al. "Decoupled Dynamic Filter Networks." In CVPR 2021.
>
>  [2] Elsayed, Gamaleldin, et al. "Revisiting spatial invariance with low-rank local connectivity." In ICML 2020.
>
>  [3] Feichtenhofer, Christoph, et al. "Slowfast networks for video recognition." In ICCV 2019.
>
>  [4] Lin, Ji, Chuang Gan, and Song Han. "Tsm: Temporal shift module for efficient video understanding." In ICCV 2019.
>
>  [5] Wang, Limin, et al. "TDN: Temporal difference networks for efficient action recognition." In CVPR 2021.
>
>  [6] Liu, Zhaoyang, et al. "Tam: Temporal adaptive module for video recognition." In ICCV 2021.
>
>  [7] Jiang, Boyuan, et al. "Stm: Spatiotemporal and motion encoding for action recognition." In ICCV 2019.

---

### Official Review · Reviewer_6Gv3 · 2021-11-07

**Correctness:** 4
**Technical Novelty And Significance:** 3
**Empirical Novelty And Significance:** 3
**Recommendation:** 6
**Confidence:** 4

**Main Review:**

Strength:
1. The paper is well-organized and easy to follow.
2. The proposed approach is simple and efficient. Temporal modeling by calibrating convolutional weights can achieve significant improvement while introducing less computation cost and additional parameters compared with other methods like (2+1)D.
3. TAdaConv can be easily plugged into any convolutional network based deep models for sequential modeling.
4. Adequate experiments on K400 and SSV2 illustrate the superiority of the proposed approach.

Weakness:
1. In Equation 7, the outputs of the first term and the second term are in different shapes. How do you add those features together? I suppose you may simply aggregate them together, but please describe the details clearly.
2. Please add the descriptions of GAPs as global average pooling over spatial dimension and GAPst as global average pooling over spatial and temporal dimension to make the audience have a good understanding of Equation 5.
3. According to Equation 4, v_t is the combination of features with temporal kernel size as 3. I would like to see more ablation study of kernel size over temporal dimension of TAdaConv (e.g., 3 vs 5 vs 7).
4. According to the recent progress on neural architecture search for action recognition [1], I would like to suggest the authors adding TadaConv into search space in the future work.

[1] Kondratyuk, Dan, et al. "Movinets: Mobile video networks for efficient video recognition." Proceedings of the IEEE/CVF Conference on Computer Vision and Pattern Recognition. 2021.

**Summary Of The Paper:**

The authors proposed a novel approach for video understanding. They present temporally adaptive convolutions (TAdaConv) based on dynamic networks by adapting the convolutional weights on each frame with its temporal contexts. The experiment results on public benchmarks demonstrate the efficiency of the proposed approach.

**Summary Of The Review:**

Overall, the paper is clear and the author propose an efficient and practical approach for video understanding. So I would like to provide a positive score in the initial stage of the review and keep tuning during the discussion period.

---

> ### Author Response · Authors · 2021-11-22
> **Response to Reviewer 6Gv3**
>
> Thank you for your acknoledgement of our writing, the effectiveness of TAdaConv and the thoroughness of the experiments. We appreciate your valuable comments, and address your questions as follows:
>
> ----
>
> 1. **Different shape in Equation 7.**
>
> Our apologies for the confusion caused. In fact, our aggregation scheme performs *strided average pooling* rather than average pooling. Hence, with a kernel size of k and appropriate padding, the output of $\text{TempAvgPool}_k(\mathbf{\tilde{x}})$ is of the same shape with $\mathbf{\tilde{x}}$. We have made this clear for all appearances of the aggregation branch.
>
> ----
>
> 2. **Description of $\text{GAP}_\text{s}$ and $\text{GAP}_\text{st} $.**
>
> Thanks, we have added the description in *Sec 3.2*.
>
> ----
>
> 3. **Ablations on kernel sizes.**
>
> In *Table A4* in the *Appendix F*, we include the ablations on the kernel size for linear/non-linear weight generation (w/o global information) as well as the ablations on the reduction ratio for the non-linear weight generation function. Related analysis is included in *Appendix F*. For convenience, we also include the results here in *Table R1* and *Table R2*. The results show that when temporal context is considered, different temopral kernel sizes have a small affect on the final performance. Hence, in the final version of our model, we still employ 3 as the kernel size as it is the most widely employed size.
>
>    ***Table R1*** Different kernel sizes for linear weight generation.
>
>    | Kernel size | 1 | 3 | 5 | 7 |
>    | ----------- | ----------- | ----------- | ----------- | ----------- |
>    | Top-1 Acc | 37.5 | 56.5 | 57.3 | 56.5 |
>
>    ***Table R2*** Different kernel sizes for non-linear weight generation.
>
>    |       | K2=1 | K2=3 | K2=5 | K2=7 |
>    | ----------- | ----------- | ----------- | ----------- | ----------- |
>    | **K1=1** | 36.8 | 57.1 | 57.8 | 57.9 |
>    | **K1=3** | 57.3 | 57.8 | 57.9 | 58.0 |
>    | **K1=5** | 57.6 | 57.9 | 58.2 | 57.9 |
>    | **K1=7** | 57.4 | 57.6 | 58.0 | 57.6 |
>
> ----
>
> 4. **Adding TAdaConv into the search space of NAS approach such as MoViNets.**
>
> Thanks for the valuable suggestion. The objective of TAdaConv is exactly to provide diversity to existing operations, and we empirically show that TAdaConv is effective for improving temporal modelling abilities for convolutional models. Indeed, MoViNets have yielded impressive results as convolutional video models. This makes the inclusion of TAdaConv in the search space of NAS approaches such as MoViNets an interesting exploration, which may provide the computer vision community with a stronger convolutional video model.

---

### Author Response · Authors · 2021-11-22
**Summary of revisions.**

Dear reviewers,

We would like to express our deepest gratitude for your meticulous examination of the paper as well as your insightful and valuable comments.
We have given careful thoughts into your suggestions and made the following revisions to our manuscript to answer your questions and address your concerns:

- **Clearer motivation description and research gap** in *Sec 1*.

    - *Problem*: temporal convolution introduces non-negligible computation on spatial convolutions.

      *Our solusion*: directly empower spatial convolution to have temporal modelling capabilities.

    - *Observation*: spatially varying dynamic filter shows stronger ability to model complex spatial contents.

      *Our hypothesis*: temporally adaptive filters can benefit complex temporal modelling.

    - *Research gap for previous dynamic filters*: (a) unable to exploit pre-trained weights; and (b) can not achieve temporal modelling.

      *Inspiration*: temporal convolution calibrates essentially spatial convolution weights before aggregation.

      *Our approach*: generates temporally adaptive weights (a) calibrated from a base weight where pre-trained weights can be loaded, and (b) calibrated dependent upon both temporal local and global context.

- **Comparison to previous dynamic filter approaches** in *Sec 3.2* and more detailedly in *Appendix J*.

    We compare our TAdaConv with the following representative approaches in both approach level and performance level.

    - mixture of experts based dynamic filter (represented by *CondConv* [1])
    - spatially-invariant dynamic filter (represented by *DynamicFilter* [2])
    - spatially-varying dynamic filter (represented by *DDF* [3])
    - dynamic filter in videos (*TAM* [4])

- **Inclusion of ReLU in revisiting temporal convolutions** in *Sec 3.1* and *Appendix A*.

  Both (2+1)D convolution with and without ReLU (details in *Appendix A*) show that temporal convolution calibrates the spatial convolution weights before aggregation.

- **Clearer description of used notations** in *Sec 3.2*.

  Added missed explanation of notation after its usage.

- **Clearer computation and parameter comparison** in *Table 2*, *Figure 3* and *Appendix B*.
  - *Table 2*: clear comparison of TAdaConv and temporal convolution. For clarity, we include example values both at operation level and network level.
  - *Figure 3*: *Performance vs. GFLOPs visualization* on SSV2
  - *Appendix B*: detailed computation analysis and comparison with other temporal modelling approaches.

- **Performance comparison of learnable calibration and dynamic calibration** in *Table 6*.
- **Performance comparison with and without our initialization** in *Table 6*.
- **Performance vs. GFLOPs** in *Figure 3*.
- **Updated performance** in *Table 8, 9, and 10*.

  Thanks to *Reviewer 4biu*, we aligned our augmentation strategy with TANet and obtained improved performance.

[1] Yang, Brandon, et al. "CondConv: conditionally parameterized convolutions for efficient inference." In NeurIPS 2019.

[2] Jia, Xu, et al. "Dynamic filter networks." In NeurIPS 2016.

[3] Zhou, Jingkai, et al. "Decoupled Dynamic Filter Networks." In CVPR 2021.

[4] Liu, Zhaoyang, et al. "Tam: Temporal adaptive module for video recognition." In ICCV 2021.

---

### Author Response · Authors · 2021-11-28
**Message for the reviewers**

Dear reviewers,

Hope this message finds you well.

We have updated the manuscript according to your comments and responded detailedly to your questions. As the discussion period will end in less than two days, we would like to kindly ask whether there is any additional concerns or questions that we might be able to address.

Thanks very much for your effort!

Best regards,

Authors

---

### Public Comment · ~Kunchang_Li1 · 2022-01-29
**Comparison with CT-Net**

Congratulations!

I appreciate the well-written paper and extensive experiments. TAda2D works well without the temporal difference, which is widely used in recent methods based on 2D CNNs.

Actually, our CT-Net[1] also design an effective module for action recognition. It also works well without extra motion information.
|Method | #Frame  | SSV2 top-1 |
|:-------------|:-------: |:-----------: |
|CT-Net [1] | 8$\times$3$\times$2|64.9 |
|TAda2D| 8$\times$3$\times$2|64.0|
|CT-Net [1] | 16$\times$3$\times$2|**65.9**|
|TAda2D| 16$\times$3$\times$2|65.6|

|Method | #Frame  | K400 top-1 |
|:-------------|:-------: |:-----------: |
|CT-Net [1] | 8$\times$3$\times$4|77.3 |
|TAda2D| 8$\times$3$\times$10|**77.4**|

The above results show that our CT-Net achieves comparable performance with TAda. Since our CT-Net has been accepted by ICLR 2021, I suggest the authors make a fair comparison with CT-Net in the camera-ready version.


[1] Li, Li, et al. "CT-Net: Channel Tensorization Network for Video Classification." In ICLR 2021.

---

### Decision · Program_Chairs · 2022-01-20

**Decision:**

Accept (Poster)

**Comment:**

The authors study the problem of video classification and propose a new module which promises to increase accuracy while keeping the computational overhead low. The main idea is not to share the spatial convolution weights over different time steps, but allow some modulation based on pooled local and global frame descriptors. The resulting module can be used as a drop-in replacement for spatial convolutions in existing models and yields competitive performance on multiple video action recognition and localisation benchmarks.

The reviewers appreciated this challenging setting and the simplicity of the main idea. They found that the paper was clearly written, well organised, and easy to follow. The reviewers raised some issues in connections with related work and the empirical evaluation which were successfully resolved during the discussion phase.

Given that computational efficiency remains as one of the most challenging topics in video understanding, I believe that this technique will be relevant for the larger video understanding community. I strongly suggest that the authors incorporate the feedback received during the discussion, especially the GFLOPS vs accuracy plots, and further clarify the relationship to existing work.